# OpenPL: Realistic Evaluation of Prompt Learning for VLM in Open Environments

## Abstract

Vision-language models (VLMs) have demonstrated impressive zero-shot capabilities across various image classification tasks. Their performance can be further enhanced through prompt learning methods. To evaluate the effectiveness of prompt learning, it is important to assess its robustness to new classes and distributional shifts. However, current studies typically assume single data distribution shifts and pre-known new class space, which still have gaps with real-world open environments where data distributions and classes are often uncertain and subject to continuous change. To better analyze the robustness of prompt learning methods in more realistic scenarios, we propose a novel evaluation benchmark called OpenPL from the following perspectives: 1) We reconstruct multiple scenarios of open environments, encompassing dynamic class changes, dynamic distribution shifts, and dynamic co-evolution of both distribution and classes; 2) We propose a series of new performance metrics for prompt learning methods based on the Dynamic Robustness Curve (DRC) to better understand their robustness in open environments; 3) We re-implement diverse prompt learning methods and evaluate their performance on the proposed OpenPL benchmark. The results show that **no current prompt learning method is robust to open environments and no meaningful performance improvement is achieved compared to the zero-shot performance**, designing robust prompt learning methods remains a difficult task. All re-implementations are available at `https://anonymous.4open.science/r/OpenPL-565E`.

## 1 Introductions

Vision-language models (VLMs) have garnered significant attention recently(Radford et al. (2021);Yao et al. (2021);Jia et al. (2021)) because of its zero-shot prediction capabilities across a wide range of visual recognition tasks. Pre-trained VLMs, such as CLIP(Radford et al. (2021)), ALIGN(Jia et al. (2021)), and BLIP(Li et al. (2022)), acquire extensive vision-language knowledge from a near-infinite number of text-image pairs available on the web. These models can be utilized directly for downstream tasks without the need for fine-tuning.

A large part of the research for VLM recently is the adaptation of pre-trained VLMs on downstream tasks(Zhang et al. (2021);Zhou et al. (2022b);Zhou et al. (2022a);Gao et al. (2024)), and prompt learning(Zhou et al. (2022b);Zhou et al. (2022a)) is a newcomer to these works, with notable improvements in VLM performance on downstream tasks and its simplicity and efficiency in design. Unlike other transfer methods in VLM, prompt learning does not rely on adding additional network layers or modifying complex network structures but rather achieves parameter-efficient VLM transfer by modifying the input text or image with some learnable text or vision prompts.

Although many prompt learning methods, such as MaPLe (Khattak et al. (2023a)) and PromptSRC (Khattak et al. (2023b)), claim to achieve strong generalization performance on downstream tasks, their evaluations often overlook more practical scenarios. Currently, most prompt learning methods depend on benchmarks established by CoCoOp (Zhou et al. (2022a)), where classes are divided into fixed base and new groups. These methods are trained solely on the base classes and tested on new classes separately. Similarly, MaPLe (Khattak et al. (2023a)) proposes benchmarks for Cross-dataset Evaluation and Domain Generalization by training on ImageNet and altering the test data distribution to assess performance across various ImageNet variants and other datasets. In open environments,

algorithms do encounter the emergence of new classes and shifts in data distributions (Zhou (2022)), but such changes are not fixed and cannot be pre-known, and sometimes both distribution and class shifts could occur simultaneously. There is still a huge gap between existing evaluation settings and realistic open environments.

**Contributions**: In this paper, we propose a novel evaluation benchmark to address the existing problems above and achieve a comprehensive evaluation for prompt learning in VLM. Firstly, we focus on the dynamic class changes and propose two scenarios based on class variations. The first scenario increases classification difficulty by continuously introducing new classes during testing. The second scenario reduces the proportion of base classes while increasing the ratio of new classes, keeping the overall label space size constant. This setup aims to measure the algorithm's robustness in an unknown and evolving label space. Secondly, we focus on dynamic distribution shifts and propose a scenario in which the testing distribution continues to change, becoming increasingly distant from the training data distribution. Thirdly, we focus on the dynamic co-evolution of both distribution and classes and propose a scenario in which both class changes and distribution shifts occur simultaneously. Moreover, we quantify the performance under these paradigms based on the Dynamic Robustness Curve (DRC), and several new performance metrics based on the DRC are proposed to help better analyze the robustness of prompt learning methods. We also re-implement different types of prompt learning methods under a unified standard and evaluate their robustness. We believe this has a positive impact on researchers in this field.

**Observations**: By researching the results of experimental evaluations on the openPL, our insights can be outlined as follows:

1. No single prompt learning method outperforms others in scenarios with dynamic classes changing, i.e., each method has its better case when new classes emerging.

2. No prompt learning methods currently show robustness to data distribution shifts, i.e., all methods suffer severe performance degradation with distribution shifts.

3. No prompt learning methods exhibit meaningful performance gains relative to the zero-shot performance with compound shifts of both distribution and label space.

4. Enhancing the model's capability for text feature extraction and class discriminability may improve robustness since the available text and image information in downstream tasks is highly imbalanced.

## 2 RELATED WORKS

**Vision-language model**    Recently, researchers have demonstrated VLM (Alayrac et al. (2022);Radford et al. (2021)), which consists of visual and textual modalities trained on large-scale image-text pairs, with strong generalization and discrimination capabilities. These VLMs like CLIP (Radford et al. (2021)), ALIGN (Jia et al. (2021)), BLIP (Li et al. (2022)), FILIP (Yao et al. (2021)), LiT (Zhai et al. (2022)) and Flamingo (Alayrac et al. (2022)) show exceptional performance across numerous visual tasks, including few-shot and zero-shot visual recognition. For example, CLIP (Radford et al. (2021)) designs objective that allows matched text representations and image representations close to each other, and learns a generalized vision-language representation on about 400M text-image pairs. Recent researches have focused on how such VLMs can be better adapted to downstream tasks by means of transfer learning (Zhou et al. (2022b);Zhou et al. (2022a);Zhang et al. (2021);Gao et al. (2024)) and knowledge distillation (Ding et al. (2022);Du et al. (2022);Gu et al. (2021)).

**Prompt learning**    The idea of prompt learning first originated in NLP(Liu et al. (2023)), a method of instructing language models to generate specific outputs by providing prompts without having to tune the pre-trained model's own parameters. Because of its parsimony and efficiency, this technique has attracted a lot of attention in the exploration of fine-tuning VLM pre-trained models to specific visual tasks downstream. For example, at the earliest time, CoOp (Zhou et al. (2022b)) explored the application of prompts to the text branch of CLIP, which was used to optimize the text embedding space of the text branch so that specific categories of the downstream task could be better adapted to the pre-trained model, while VPT (Jia et al. (2022)) later provided a solution for introducing prompts into the visual encoder. After that CoCoOp (Zhou et al. (2022a)) explores the introduction of image information into text prompts to make the unified context into an instance-adaptive context. Moreover

MaPLe (Khattak et al. (2023a)) experiment with the co-optimization of prompts for both the textual and visual sides of the prompts. The aforementioned methods are all dedicated to improving the form of prompts and enhancing the model's ability to extract image features. Different from the above methods, ProGrad (Zhu et al. (2023)) and KgCoOp (Yao et al. (2023b)) explore how to improve the generalization ability of pre-trained models by better preserving their knowledge. ProGrad only updates prompts whose gradients do not conflict with the knowledge of the pre-trained model to prevent general knowledge from being forgotten; KgCoOp uses the gap from the fine-tuned prompts to general knowledge as a regularization term for constrained models. Similarly, RPO (Lee et al. (2023)) leverages masked attention to prevent the internal representation shift in the pre-trained model to reduce the decline in the generalization ability of the pre-trained model. Moreover, TCP (Yao et al. (2023a)) and ProDA (Lu et al. (2022)) choose to enhance the textual side of the representation, with TCP incorporating prior knowledge about classes to enhance the discriminability of classes; ProDA learns output embeddings of textual prompts rather than input embeddings. PromptSRC (Khattak et al. (2023b)) achieves notable performance by simultaneously improving prompt learning from three aspects: preventing pre-trained knowledge forgetting; prompt ensemble; and increasing text diversity to mitigate sample diversity.

In the following, we will introduce the evaluation paradigms, performance metrics, the robustness definitions of prompt learning, benchmark results, and conclusions.

## 3 EVALUATION PARADIGMS

In previous experiments on prompt learning, changes in classes and data distributions during testing were fixed, lacking dynamism. To address this, our benchmark introduces dynamic scenarios where new classes emerge, data distributions shift, and both distributions and classes co-evolve. This enables a more comprehensive analysis of the robustness of existing prompt learning algorithms in such dynamic environments.

### 3.1 DYNAMIC CLASSES CHANGES

In this paradigm, we introduce two scenarios in open environments where new classes continuously emerge. For each experiment, we randomly select half of the classes from a single dataset to serve as base classes, while the remaining half are designated as new classes. The algorithms are trained exclusively on samples from the base classes. During testing, they encounter a dynamic situation in which both base and new classes co-exist, with their quantities continuously changing.

**Emerging New Classes**  In this scenario, during testing, the base classes remain consistently present, while new classes continuously emerge. As new classes are introduced, the algorithm faces a larger class space, making the classification task more challenging. We define the class changing level $t$ as the ratio of new classes in the test set relative to the base classes, where a higher $t$ signifies a more complex classification task. We ensure that as $t$ increases, the new classes from the previous level are subsets of the corresponding groups for the next level.

**Varying Ratio of New Classes**  In this scenario, during testing, the base classes do not remain constant; instead, they decrease as the number of new classes increases. The size of the class space remains unchanged, while the quantities of new and base classes vary synchronously. To achieve this, we define the class changing level $t$ to represent the proportion of new classes in the test set relative to all classes. As $t$ increases, new classes continually emerge while base classes diminish. To ensure comparability of performance across different levels of inconsistency, we ensure that as $t$ increases, the new classes from the previous level are subsets of the corresponding groups for the next level, and the removed base classes are subsets of their respective groups from the previous level.

### 3.2 DYNAMIC DISTRIBUTION SHIFTS

Under this paradigm, we train on all classes of ImageNet and test on a mixture of ImageNet and its variants with continuously changing proportions. We define a distribution change level $t$ to represent the proportion of samples from the ImageNet variants within the entire test set. As the value of $t$ increases, the proportion of data from ImageNet decreases, while the proportion of data from the

variants gradually increases. At the same time, we ensure that the class space of the training set remains consistent with the variants. As $t$ increases, the samples from the ImageNet variants at the previous level are subsets of those at the subsequent level, with the reduced ImageNet samples coming from the previous level.

### 3.3 DYNAMIC CO-EVOLUTION OF DISTRIBUTION AND CLASS VARIATION

In this paradigm, we train on ImageNet and test on a mixed dataset that combines ImageNet and other datasets. The number of classes and samples belonging to ImageNet in the test set is kept equal to those from other datasets, with classes and samples randomly selected. We define a class and distribution change level $t$ to represent the proportion of cross-dataset samples among all test samples. As $t$ increases, the proportion of samples from the other datasets also increases. As the value of $t$ increases, the samples and classes from the cross-dataset continuously increase, while the samples and classes from ImageNet steadily decrease. We ensure that as $t$ increases, the samples from other datasets at the previous level are subsets of those at the subsequent level, with the reduced ImageNet samples coming from the previous level.

## 4 PERFORMANCE METRICS

To achieve a fair and comprehensive evaluation, for the four evaluation paradigms designed above we introduce a comprehensive set of evaluation metrics to analyze the robustness of prompt learning in VLM. We first define the model accuracy at inconsistency $t$ as $Acc(t)$. Zero-shot pre-trained CLIP also has $Acc_{zs}(t)$ at any $t$, which represents only the accuracy obtained from the same test set at $t$, for comparing the performance improvements from other prompt learning methods on the pre-trained model. The accuracy of the model in changing environments is mapped to a function of inconsistency $t$ of different scenarios. In this way we construct the Dynamic Robustness Curve (DRC) and propose several metrics based on it including 1) Area Under the Curve (AUC) which analyzes the overall robustness of the model; 2) Worst-Case Accuracy (WA) which represents the worst performance in open environments; 3) Expected Variation Magnitude (EVM) measuring the overall magnitude of the change in accuracy; 4) Variation Stability (VS) quantifying the stability of variation magnitude; 5) Positive Area (PA) measuring the performance gain in parts where the algorithm surpasses zero-shot performance; 6) Negative Area (NA) measuring the performance degradation in parts where the algorithm underperforms compared to the zero-shot performance. Table 1 provides a detailed formulation of these metrics.

Table 1: The Definition of Performance Metrics

| Metrics | Formulation |
| --- | --- |
| Area Under the Curve (AUC) | $\int_0^1 Acc(t)dt$ |
| Worst-Case Accuracy (WA) | $min_{t \in [0,1]} Acc(t)$ |
| Expected Variation Magnitude (EVM) | $\int_0^1 \lvert Acc'(t) \rvert dt$ |
| Variation Stability (VS) | $\int_0^1 (Acc'(t) - \int_0^1 Acc'(t)dt)^2 dt$ |
| Positive Area (PA) | $\int_{t \in D} Acc(t) - Acc_{zs}(t)dt \; D = \{x \mid Acc(t) \geq Acc_{zs}(t)\}$ |
| Negative Area (NA) | $\int_{t \in D} Acc_{zs} - Acc(t)(t)dt \; D = \{x \mid Acc(t) < Acc_{zs}(t)\}$ |

Moreover, in order to fairly compare the performance of different methods in scenarios with changing classes, we use Friedman rank (Friedman (1937); Friedman (1940)) to get the average ranks of these methods across different scenarios and different datasets.

$$rank_F = \frac{1}{m} \sum_{i=1}^{m} rank_i$$

We count the average ranks at the 6 $t$-values settings for each scenario, where $m = 6$, and the overall average ranks across $n$ datasets where $m = 6 \times n$. $rank_i$ is the rank of a prompt learning method in the $i$-th setting. Additionally, we will re-rank the methods to determine the final rank based on the results of the Friedman ranking.

Based on the proposed performance metrics, we further define robust prompt learning in open environments to enhance our understanding of the robustness gain of prompt learning methods compared to zero-shot performance. This includes the concepts of performance-gain robustness and decay-gain-ratio robustness.

**Definition (Performance-Gain Robustness)**   We define the AUC obtained from the VLM's zero-shot prediction as $AUC_{zs}$. A prompt learning method $A$ in VLM returns a model that can be tested with any class and distribution change level $t$. If there exists $\delta_{AUC}$ such that $AUC - AUC_{zs} \geq \delta_{AUC}$ holds for all $t$, we say $A$ achieves $\delta_{AUC}$-performance-gain robustness.

**Definition (Decay-Gain-Ratio Robustness)**   A prompt learning method $A$ in VLM returns a model that can be tested with any class and distribution change level $t$. If there exists $\delta_{PN}$ such that $PA - NA \leq \delta_{PN}$ holds for all $t$, we say $A$ exhibits $\delta_{PN}$-decay-gain-ratio robustness.

## 5 BENCHMARK RESULTS

### 5.1 EXPERIMENT SETUP

**Methods**   In our experiments, we evaluate 11 prompt learning methods based on the pre-trained CLIP using a Vision Transformer (ViT). The methods are as follows: text-based prompt learning methods CoOp, CoCoOp, ProGrad, KgCoOp, TCP, ProDA, RPO; the visual prompt learning VPT; the text-vision prompt learning methods MaPLe and PromptSRC. We also evaluated the zero-shot prediction capability of the pre-trained model CLIP as a baseline in order to compare the performance of these prompt learning methods.

**Datasets**   Following CoOp and CoCoOp, we evaluate the performance of these prompt learning methods on 11 diverse image classification datasets that cover a variety of recognition tasks. These datasets include: two generic object datasets, ImageNet (Deng et al. (2009)) and Caltech101 (Fei-Fei et al. (2004)); one texture dataset DTD (Cimpoi et al. (2014)); a satellite image dataset EuroSAT (Helber et al. (2019)); five fine-gained dataset FGVCAircraft (Maji et al. (2013)), Food101 (Bossard et al. (2014)), Flowers102 (Nilsback & Zisserman (2008)), OxfordPets (Parkhi et al. (2012)), StanfordCars (Krause et al. (2013)); one scene recognition dataset SUN397 (Xiao et al. (2010)), and an action recognition dataset UCF101 (Soomro et al. (2012)). In the Dynamic Distribution shifts paradigm, we utilize four variants of ImageNet including ImageNetV2 (Recht et al. (2019)), ImageNetSketch (Wang et al. (2019)), ImageNet-A (Hendrycks et al. (2021b)), ImageNet-R (Hendrycks et al. (2021a)).

**Implementation Details**   For all experiments, we adopted a unified parameter setting to ensure a fair comparison. In the two scenarios of Dynamic Classes Changes, we set the learning rate $\eta$ as $2 \times 10^{-3}$, and the total number of epochs is 50 for each dataset. And for mixed proportions of ImageNet variants and the Dynamic Co-evolution of Distribution and Class Variation paradigms, the training on ImageNet is set to run for 10 epochs. Additionally, to ensure a balanced mixture of different datasets, we cap the maximum number of classes and the sample size for all datasets during testing to maintain equivalence. We set the length of text or vision prompts in all methods as 4. We sample 16 samples per class from the training dataset and test all methods on the full test dataset, following the commonly used few-shot evaluation protocol as that in CLIP. We adopt ViT-Base/16 as the backbone network for all experiments. The initial setting for text prompts is fixed to "X X X X" the initialization of vision prompts follows a zero-mean Gaussian distribution with a standard deviation of 0.02. For a fair comparison, the final results were averaged over three rounds of experiments. To plot the curve DAC, we sampled six points for t as 0, 0.2, 0.4, 0.6, 0.8, 1.0. To ensure reliability, the label space is randomized for each round of training and testing, and we conduct experiments three times with seed values of 1, 2, and 3. The results for each sampling point were averaged over three experiments, and linear interpolation was used for the other points. Our experiments are conducted on NVIDIA A800 GPUs. The complete experimental results are presented in the Appendix A.

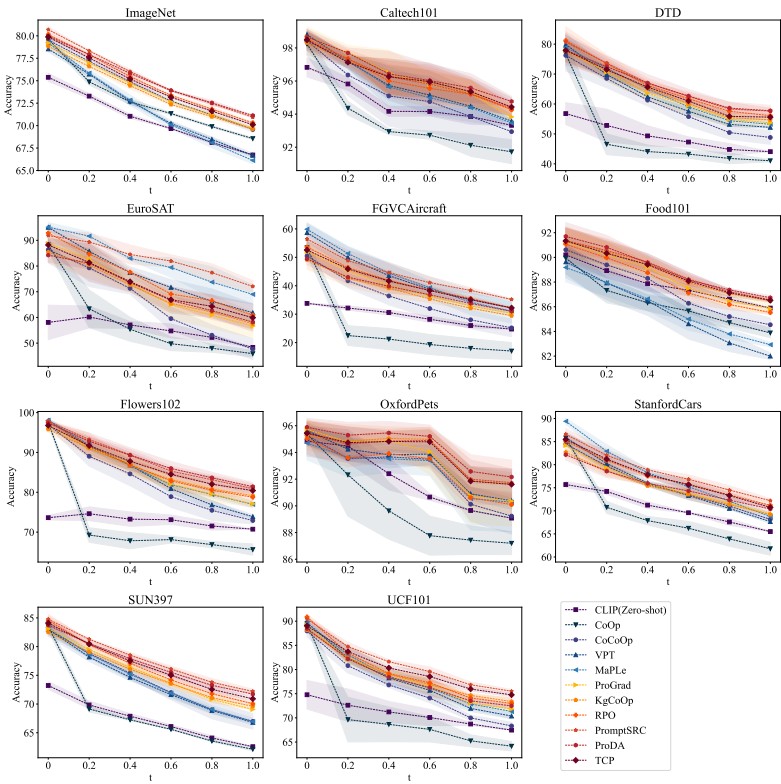

Figure 1: Results of prompt learning methods under emerging new classes on 11 datasets. As the value of $t$ increases, the number of base classes remains constant, while the number of new classes gradually increases until it equals the number of base classes.

## 5.2 Prompt Learning under Dynamic Classes Changes

> **Observation**
>
> **For dynamic class changes, it is challenging for any prompt learning method to consistently achieve optimal performance across different datasets.**

**Prompt Learning under Emerging New Classes** In Figure 1, we present a comparison of different prompt learning methods under emerging new classes, along with their corresponding outcomes. It can be observed that as more new classes are introduced during testing, the performance of various prompt learning methods generally shows a gradually declining trend. However, the speed of this decline varies between different methods, leading to changes in the relative performance of these methods as new classes continue to emerge. For example, in the FGVCAircraft dataset, the performance of PromptSRC initially falls significantly behind that of MaPLe and VPT. However, as the classification pressure increases, the performance of PromptSRC begins to surpass these methods and maintains the lead among all the methods. **This indicates that, it is challenging for any single method to consistently maintain optimal performance across different open environments.**

In Table 2, we present the evaluation results of the metrics on DTD under emerging new classes. We can see that, under these evaluation paradigms, metrics like $Acc(0)$, AUC, EVM, and VS are often not necessarily related. A method with a higher $Acc(0)$ may even perform worse than CLIP in other metrics. And it's worth noting that CLIP often surpasses most methods in terms of EVM.

**Prompt Learning under Varying Ratio of New Classes** In Figure 2, we compare various prompts learning methods under varying ratio of new classes and get the results. We can find that the

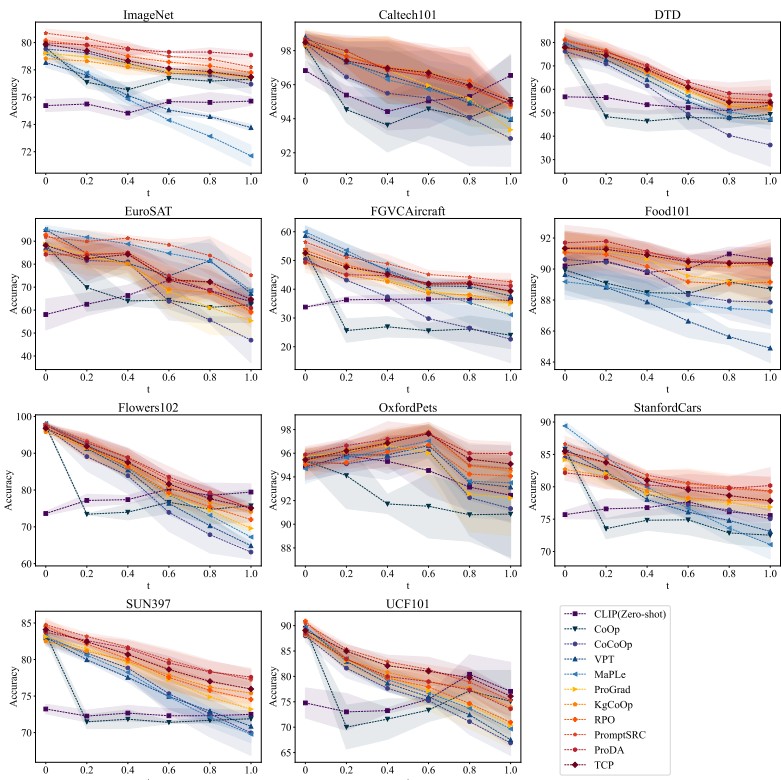

Figure 2: Results of prompt learning methods under varying ratio of new classes on 11 datasets. As the value of $t$ increases, the number of base classes gradually decreases while the number of new classes gradually increases. Always keep the total number of classes constant.

Table 2: Evaluation on DTD under Emerging New Classes. In the table, higher values for the metrics Acc(0), AUC, WA, and PA are better, while lower values for EVM, VS, and NA are preferable.

| Dataset | Methods | Acc(0) | AUC | WA | EVM | VS | PA | NA |
|---------|---------|--------|-----|-----|-----|-----|-----|-----|
| | CLIP(Zero-shot) | 0.568 | 0.489 | 0.441 | **0.127** | **0.003** | / | / |
| | CoOp | 0.767 | 0.469 | 0.410 | 0.357 | 0.333 | 0.015 | 0.032 |
| | CoCoOp | 0.762 | 0.597 | 0.488 | 0.274 | 0.011 | 0.108 | **0.000** |
| | VPT | 0.799 | 0.618 | 0.522 | 0.277 | 0.023 | 0.129 | **0.000** |
| | MaPLe | 0.789 | 0.633 | 0.532 | 0.257 | 0.011 | 0.144 | **0.000** |
| DTD | ProGrad | 0.779 | 0.629 | 0.537 | 0.242 | 0.012 | 0.140 | **0.000** |
| | KgCoOp | 0.777 | 0.637 | 0.547 | 0.230 | 0.010 | 0.147 | **0.000** |
| | RPO | **0.812** | 0.655 | 0.561 | 0.251 | 0.013 | 0.166 | **0.000** |
| | PromptSRC | 0.807 | **0.661** | 0.576 | 0.231 | 0.013 | **0.172** | **0.000** |
| | ProDA | 0.779 | 0.657 | **0.577** | 0.202 | 0.007 | 0.168 | **0.000** |
| | TCP | 0.779 | 0.642 | 0.556 | 0.223 | 0.012 | 0.152 | **0.000** |

accuracy of the model is not always monotonically declining as new classes increases and base classes decreases, but instead varying performance changes occur under different ratios of new classes to base classes. For example, in the EuroSAT dataset, the performance of VPT and TCP does not always decrease with the increasing ratio of new classes to base classes; There are instances where the performance actually improves with more new classes and fewer base classes. In real-world scenarios where the label space is unknown, it's not always the case that having fewer new classes and more base classes will lead to better performance. **Unknown mixed class proportions can sometimes actually be more detrimental to the performance than having more new classes. Moreover, on each dataset, there does not exist a certain method that can be optimal under various ratios of new classes**, e.g., on the Stanford-cars dataset, the best in terms of accuracy goes from MaPLe to PromptSRC to ProDA as the value of t increases.

Table 3: Evaluation on FGVCAircraft under Varying Ratio of New Classes

| Dataset | Methods | Acc(0) | AUC | WA | EVM | VS | PA | NA |
|---|---|---|---|---|---|---|---|---|
| | CLIP(Zero-shot) | 0.338 | 0.362 | 0.338 | **0.034** | **0.003** | / | / |
| | CoOp | 0.534 | 0.286 | 0.240 | 0.332 | 0.303 | 0.013 | 0.081 |
| | CoCoOp | 0.505 | 0.347 | 0.226 | 0.279 | 0.008 | 0.031 | 0.034 |
| | VPT | 0.587 | 0.459 | 0.376 | 0.210 | 0.012 | 0.097 | **0.000** |
| | MaPLe | **0.599** | 0.440 | 0.311 | 0.288 | 0.005 | 0.085 | 0.006 |
| FGVCAircraft | ProGrad | 0.520 | 0.423 | 0.350 | 0.170 | **0.003** | 0.062 | 0.001 |
| | KgCoOp | 0.492 | 0.413 | 0.356 | 0.136 | 0.005 | 0.051 | **0.000** |
| | RPO | 0.536 | 0.447 | 0.392 | 0.155 | 0.009 | 0.085 | **0.000** |
| | PromptSRC | 0.564 | **0.478** | **0.426** | 0.139 | 0.006 | **0.116** | **0.000** |
| | ProDA | 0.495 | 0.439 | 0.411 | 0.088 | 0.007 | 0.077 | **0.000** |
| | TCP | 0.526 | 0.445 | 0.394 | 0.132 | 0.006 | 0.084 | **0.000** |

As shown in Table 3, the performance variability and stability of the algorithms under the Varying Ratio of New Classes paradigm on FGVCAircraft are worse than that of CLIP's zero-shot predictions. Additionally, no algorithm has managed to maintain excellent performance on base classes while also demonstrating a slower and more stable decline in performance.

## 5.3 PROMPT LEARNING UNDER DYNAMIC DISTRIBUTION SHIFTS

> **Observation**
>
> **There is no significant improvement across algorithms when addressing the issue of dynamic data distribution shifts.**

As shown in Figure 3, we can clearly observe that for different ImageNet variants, the performance decline of various prompt learning methods, as the proportion of variant data increases, almost mirrors the zero-shot predictions of CLIP. As indicated in Table 4, the metrics representing the degree of change and stability, such as EVM and VS, show minimal differences. Prompt learning does not demonstrate strong performance when confronted with changes in data distribution. The slight performance gains on varying data distributions are primarily attributed to improved fine-tuning on ImageNet, but they do little to mitigate the performance degradation trend in the Dynamic Distribution shifts paradigm.

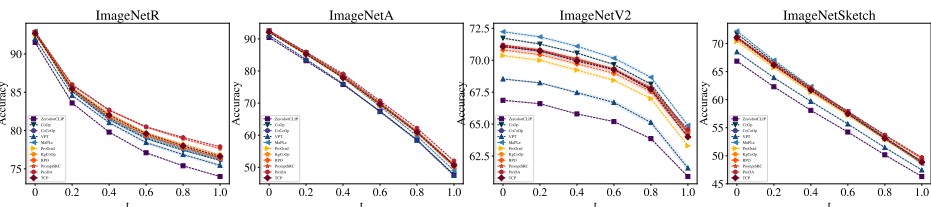

Figure 3: Results of prompt learning methods under Dynamic Distribution shifts.

Table 4: Evaluation on ImageNet-R under Dynamic Distribution shifts

| Dataset | Methods | Acc(0) | AUC | WA | EVM | VS | PA | NA |
|---|---|---|---|---|---|---|---|---|
| | CLIP(Zero-shot) | 0.915 | 0.797 | 0.740 | 0.175 | 0.014 | / | / |
| | CoOp | 0.927 | 0.815 | 0.761 | 0.166 | 0.013 | 0.017 | **0.000** |
| | CoCoOp | 0.928 | 0.818 | 0.764 | 0.164 | 0.013 | 0.020 | **0.000** |
| | VPT | 0.920 | 0.809 | 0.755 | 0.166 | 0.012 | 0.012 | **0.000** |
| | MaPLe | **0.930** | 0.816 | 0.762 | 0.168 | 0.013 | 0.019 | **0.000** |
| ImageNet-R | ProGrad | 0.925 | 0.818 | 0.766 | 0.159 | 0.011 | 0.021 | **0.000** |
| | KgCoOp | 0.929 | 0.821 | 0.768 | 0.160 | 0.012 | 0.024 | **0.000** |
| | RPO | 0.928 | 0.818 | 0.763 | 0.165 | 0.012 | 0.020 | **0.000** |
| | PromptSR | 0.928 | 0.826 | 0.776 | 0.152 | **0.010** | 0.029 | **0.000** |
| | ProDA | 0.929 | **0.827** | **0.779** | **0.150** | 0.011 | **0.030** | **0.000** |
| | TCP | 0.927 | 0.819 | 0.766 | 0.160 | 0.011 | 0.022 | **0.000** |

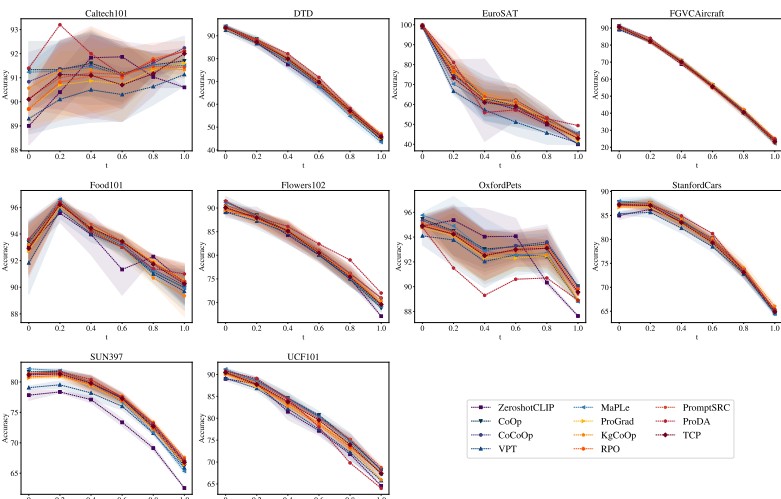

Figure 4: Results of prompt learning methods cross various datasets.

Table 5: Evaluation Metrics under Dynamic Co-evolution of Distribution and Class Variation

| Dataset | Methods | Acc(0) | AUC | WA | EVM | VS | PA | NA |
|---------|---------|--------|-----|-----|-----|-----|-----|-----|
| | CLIP(Zero-shot) | 0.986 | 0.629 | 0.400 | 0.586 | 0.100 | / | / |
| | CoOp | 0.995 | 0.650 | 0.451 | 0.544 | 0.104 | 0.021 | **0.000** |
| | CoCoOp | 0.992 | 0.649 | 0.445 | 0.547 | 0.151 | 0.022 | 0.001 |
| | VPT | 0.991 | 0.581 | 0.407 | 0.584 | 0.275 | 0.000 | 0.023 |
| | MaPLe | 0.994 | 0.629 | 0.458 | 0.537 | 0.225 | 0.012 | 0.001 |
| EuroSAT | ProGrad | 0.992 | 0.643 | 0.417 | 0.575 | 0.114 | 0.015 | **0.000** |
| | KgCoOp | 0.992 | 0.657 | 0.437 | 0.555 | **0.094** | 0.028 | **0.000** |
| | RPO | 0.994 | 0.640 | 0.431 | 0.563 | 0.124 | 0.012 | **0.000** |
| | PromptSRC | 0.994 | **0.658** | 0.449 | 0.545 | **0.094** | **0.029** | **0.000** |
| | ProDA | **1.000** | 0.644 | **0.494** | **0.534** | 0.257 | 0.025 | 0.003 |
| | TCP | 0.994 | 0.630 | 0.429 | 0.564 | 0.163 | 0.007 | 0.001 |

## 5.4 PROMPT LEARNING UNDER DYNAMIC CO-EVOLUTION OF DISTRIBUTION AND CLASS VARIATION

> **Observation**
>
> **Prompt learning exhibits performance nearly on par with the zero-shot prediction capabilities of CLIP, showing little to no improvement in scenarios characterized by the coupling of dynamic distribution and class changes.**

As shown in Figure 4, for relatively simple classification datasets like Caltech101 and OxfordPets, the performance variations under different cross-dataset ratios are minimal. **However, for datasets where CLIP already exhibits significant performance degradation across datasets, the various algorithms face similar challenges, as observed in the Dynamic Distribution Shifts paradigm, showing no notable performance improvements compared to CLIP's zero-shot predictions.** Furthermore, as illustrated in Figure 5, the differences in metrics such as AUC and EVM between the algorithms and zero-shot predictions under EuroSAT are minor, while the stability of algorithm performance, as reflected in the VS metric, generally shows varying degrees of decline.

## 5.5 ROBUSTNESS ANALYSIS OF PROMPT LEARNING METHODS

In Tables 6 9, we compare the robustness of different algorithms across various dynamic scenarios. Tables 6 and 7 reveal that CoOp exhibits the poorest robustness when facing various class changes, as it lacks any robustness handling mechanisms as an initial prompt learning method. Similarly, CoCoOp, VPT, and MaPLe demonstrate comparable poor performance. While these methods continuously improve prompt formulation and somewhat enhance the model's ability to extract image features,

Table 6: Average Robustness and Ranks under Emerging New Classes.

| Methods | $\delta_{AUC}$ | $\delta_{PN}$ | Friedman rank | Final rank |
|---|---|---|---|---|
| CLIP(Zero-shot) | / | / | 9.788 | 10 |
| CoOp | -0.013 | -0.012 | 9.939 | 11 |
| CoCoOp | 0.053 | 0.053 | 8.212 | 9 |
| VPT | 0.071 | 0.071 | 6.636 | 8 |
| MaPLe | 0.081 | 0.081 | 6.136 | 7 |
| ProGrad | 0.072 | 0.072 | 5.742 | 5 |
| KgCoOp | 0.072 | 0.072 | 5.909 | 6 |
| RPO | 0.083 | 0.083 | 4.227 | 4 |
| PromptSRC | **0.102** | **0.102** | 1.955 | 1 |
| ProDA | 0.081 | 0.081 | 3.500 | 2 |
| TCP | 0.083 | 0.083 | 3.955 | 3 |

Table 7: Average Robustness and Ranks under Varying Ratio of New Classes.

| Methods | $\delta_{AUC}$ | $\delta_{PN}$ | Friedman rank | Final rank |
|---|---|---|---|---|
| CLIP(Zero-shot) | / | / | 8.152 | 9 |
| CoOp | -0.014 | -0.012 | 8.970 | 11 |
| CoCoOp | 0.013 | 0.016 | 8.742 | 10 |
| VPT | 0.037 | 0.039 | 7.288 | 8 |
| MaPLe | 0.045 | 0.046 | 6.879 | 7 |
| ProGrad | 0.040 | 0.041 | 6.379 | 6 |
| KgCoOp | 0.045 | 0.046 | 5.758 | 5 |
| RPO | 0.053 | 0.054 | 4.455 | 4 |
| PromptSRC | **0.074** | **0.074** | 2.212 | 1 |
| ProDA | 0.059 | 0.059 | 3.273 | 2 |
| TCP | 0.057 | 0.057 | 3.894 | 3 |

Table 8: Average Robustness and Ranks under Dynamic Distribution shifts.

| Methods | $\delta_{AUC}$ | $\delta_{PN}$ | Friedman rank | Final rank |
|---|---|---|---|---|
| CLIP(Zero-shot) | / | / | 10.875 | 11 |
| CoOp | 0.031 | 0.031 | 4.625 | 5 |
| CoCoOp | 0.030 | 0.030 | 5.208 | 6 |
| VPT | 0.011 | 0.011 | 10.125 | 10 |
| MaPLe | 0.031 | 0.031 | 4.583 | 4 |
| ProGrad | 0.026 | 0.026 | 7.917 | 9 |
| KgCoOp | 0.032 | 0.032 | 3.375 | 2 |
| RPO | 0.028 | 0.028 | 7.250 | 8 |
| PromptSRC | 0.033 | 0.033 | 3.917 | 3 |
| ProDA | **0.035** | **0.035** | 1.958 | 1 |
| TCP | 0.029 | 0.029 | 6.167 | 7 |

Table 9: Average Robustness and Ranks under Dynamic Co-evolution of Distribution and Class Variation.

| Methods | $\delta_{AUC}$ | $\delta_{PN}$ | Friedman rank | Final rank |
|---|---|---|---|---|
| CLIP(Zero-shot) | / | / | 8.400 | 10 |
| CoOp | 0.012 | 0.013 | 4.400 | 4 |
| CoCoOp | 0.012 | 0.013 | 4.267 | 2 |
| VPT | -0.004 | 0.000 | 9.450 | 11 |
| MaPLe | 0.007 | 0.009 | 5.833 | 6 |
| ProGrad | 0.007 | 0.008 | 7.450 | 9 |
| KgCoOp | **0.013** | 0.013 | 4.467 | 5 |
| RPO | 0.008 | 0.008 | 6.667 | 8 |
| PromptSRC | 0.013 | **0.014** | 4.200 | 1 |
| ProDA | 0.011 | 0.013 | 4.383 | 3 |
| TCP | 0.008 | 0.009 | 6.483 | 7 |

they fail to consider generalization to new classes and do not specifically address robustness in downstream tasks. In contrast, methods such as RPO, ProDA, TCP, and PromptSRC, which aim to enhance the model's ability to extract and differentiate text features, achieve better robustness in these scenarios and consistently rank highly in overall performance. **We contend that improving the model's capability to extract text features and distinguish between classes contributes to generalization in real-world scenarios with unknown label spaces.**

In Table 8, we observe that under the Dynamic Distribution Shifts scenario, the robustness of most methods shows little variation, except for VPT, which significantly underperforms compared to the others. In this paradigm, algorithms like TCP and RPO, which perform well under class changes, do not guarantee similarly strong performance; conversely, the earlier methods CoOp and CoCoOp actually perform better than these newer approaches.

In Table 9, under the Dynamic Co-evolution of Distribution and Class Variation paradigm, ProDA and PromptSRC continue to demonstrate good robustness and high rankings. Aside from VPT, the earlier algorithms CoOp and CoCoOp increasingly outperform other methods. This indicates that most algorithms fundamentally lack the capability to effectively address the challenges posed by cross-dataset variation.

## 6 CONCLUSION

Research on robust prompt learning is an important step toward more practical tasks of VLM. This paper provides a new benchmark to evaluate the robustness of prompt learning in open environments, which includes dynamic class changes, dynamic distribution shifts, and dynamic co-evolution of both distributions and classes. We present several new performance metrics to help analyze the robustness and conduct experiments on commonly adopted prompt learning methods. The results reveal that current prompt learning methods in VLMs are not robust to class and data distribution changes. On the contrary, they highly rely on the zero-shot ability of CLIP and show no significant robust improvement compared to baseline zero-shot performance. Of course, the issues models face in real-world environments may be more complex than the paradigms we have proposed. We hope that our work can help promote the study of prompt learning in real-world scenarios.

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

# A    APPENDIX

The complete detailed experimental results are presented as follow, including the evaluation under those paradigms; and the robustness and ranks of prompt learning methods on 11 datasets under those paradigms.

Table 10: Evaluation on ImageNet under Emerging New Classes

| Dataset | Methods | Acc(0) | AUC | WA | EVM | VS | PA | NA |
|---|---|---|---|---|---|---|---|---|
| ImageNet | CLIP(Zero-shot) | 0.754 | 0.706 | 0.667 | **0.087** | **0.000** | / | / |
| | CoOp | 0.796 | 0.726 | 0.686 | 0.111 | 0.004 | 0.019 | **0.000** |
| | CoCoOp | 0.795 | 0.741 | 0.696 | 0.100 | **0.000** | 0.035 | **0.000** |
| | VPT | 0.785 | 0.720 | 0.667 | 0.118 | 0.001 | 0.013 | **0.000** |
| | MaPLe | 0.792 | 0.719 | 0.661 | 0.131 | 0.001 | 0.013 | 0.001 |
| | ProGrad | 0.792 | 0.740 | 0.697 | 0.095 | **0.000** | 0.034 | **0.000** |
| | KgCoOp | 0.788 | 0.737 | 0.696 | 0.092 | **0.000** | 0.031 | **0.000** |
| | RPO | 0.801 | 0.748 | 0.703 | 0.098 | **0.000** | 0.041 | **0.000** |
| | PromptSRC | **0.807** | **0.753** | 0.709 | 0.098 | **0.000** | **0.047** | **0.000** |
| | ProDA | 0.800 | 0.751 | **0.711** | 0.089 | **0.000** | 0.045 | **0.000** |
| | TCP | 0.799 | 0.745 | 0.701 | 0.098 | **0.000** | 0.039 | **0.000** |

Table 11: Evaluation on Caltech101 under Emerging New Classes

| Dataset | Methods | Acc(0) | AUC | WA | EVM | VS | PA | NA |
|---|---|---|---|---|---|---|---|---|
| Caltech101 | CLIP(Zero-shot) | 0.968 | 0.946 | 0.933 | **0.035** | 0.001 | / | / |
| | CoOp | 0.982 | 0.934 | 0.917 | 0.065 | 0.005 | 0.001 | 0.011 |
| | CoCoOp | 0.984 | 0.952 | 0.929 | 0.055 | 0.001 | 0.006 | **0.000** |
| | VPT | **0.988** | 0.957 | 0.936 | 0.052 | **0.000** | 0.011 | **0.000** |
| | MaPLe | 0.984 | 0.956 | 0.935 | 0.050 | **0.000** | 0.010 | **0.000** |
| | ProGrad | 0.986 | 0.963 | 0.938 | 0.048 | **0.000** | 0.017 | **0.000** |
| | KgCoOp | 0.983 | 0.962 | 0.943 | 0.040 | **0.000** | 0.016 | **0.000** |
| | RPO | 0.985 | 0.961 | 0.943 | 0.043 | **0.000** | 0.015 | **0.000** |
| | PromptSRC | 0.984 | 0.962 | 0.944 | 0.041 | **0.000** | 0.016 | **0.000** |
| | ProDA | 0.987 | **0.965** | **0.948** | 0.039 | **0.000** | **0.019** | **0.000** |
| | TCP | 0.985 | 0.962 | 0.944 | 0.041 | **0.000** | 0.016 | **0.000** |

Table 12: Evaluation on DTD under Emerging New Classes

| Dataset | Methods | Acc(0) | AUC | WA | EVM | VS | PA | NA |
|---|---|---|---|---|---|---|---|---|
| DTD | CLIP(Zero-shot) | 0.568 | 0.489 | 0.441 | **0.127** | **0.003** | / | / |
| | CoOp | 0.767 | 0.469 | 0.410 | 0.357 | 0.333 | 0.015 | 0.032 |
| | CoCoOp | 0.762 | 0.597 | 0.488 | 0.274 | 0.011 | 0.108 | **0.000** |
| | VPT | 0.799 | 0.618 | 0.522 | 0.277 | 0.023 | 0.129 | **0.000** |
| | MaPLe | 0.789 | 0.633 | 0.532 | 0.257 | 0.011 | 0.144 | **0.000** |
| | ProGrad | 0.779 | 0.629 | 0.537 | 0.242 | 0.012 | 0.140 | **0.000** |
| | KgCoOp | 0.777 | 0.637 | 0.547 | 0.230 | 0.010 | 0.147 | **0.000** |
| | RPO | **0.812** | 0.655 | 0.561 | 0.251 | 0.013 | 0.166 | **0.000** |
| | PromptSRC | 0.807 | **0.661** | 0.576 | 0.231 | 0.013 | **0.172** | **0.000** |
| | ProDA | 0.779 | 0.657 | **0.577** | 0.202 | 0.007 | 0.168 | **0.000** |
| | TCP | 0.779 | 0.642 | 0.556 | 0.223 | 0.012 | 0.152 | **0.000** |

Table 13: Evaluation on EuroSAT under Emerging New Classes

| Dataset | Methods | Acc(0) | AUC | WA | EVM | VS | PA | NA |
|---------|---------|--------|-----|-----|-----|-----|-----|-----|
| | CLIP(Zero-shot) | 0.581 | 0.555 | 0.484 | **0.139** | 0.011 | / | / |
| | CoOp | 0.882 | 0.567 | 0.459 | 0.423 | 0.182 | 0.035 | 0.022 |
| | CoCoOp | 0.863 | 0.660 | 0.477 | 0.386 | 0.012 | 0.106 | **0.000** |
| | VPT | 0.950 | 0.759 | 0.618 | 0.332 | 0.008 | 0.204 | **0.000** |
| | MaPLe | **0.951** | 0.820 | 0.690 | 0.262 | 0.009 | 0.265 | **0.000** |
| EuroSAT | ProGrad | 0.887 | 0.709 | 0.567 | 0.320 | 0.012 | 0.154 | **0.000** |
| | KgCoOp | 0.854 | 0.704 | 0.575 | 0.279 | 0.007 | 0.149 | **0.000** |
| | RPO | 0.928 | 0.749 | 0.609 | 0.319 | 0.011 | 0.194 | **0.000** |
| | PromptSRC | 0.919 | **0.830** | **0.721** | 0.198 | **0.004** | **0.275** | **0.000** |
| | ProDA | 0.842 | 0.710 | 0.582 | 0.260 | 0.007 | 0.155 | **0.000** |
| | TCP | 0.882 | 0.721 | 0.599 | 0.283 | 0.009 | 0.166 | **0.000** |

Table 14: Evaluation on FGVCAircraft under Emerging New Classes

| Dataset | Methods | Acc(0) | AUC | WA | EVM | VS | PA | NA |
|---------|---------|--------|-----|-----|-----|-----|-----|-----|
| | CLIP(Zero-shot) | 0.338 | 0.292 | 0.247 | **0.091** | **0.000** | / | / |
| | CoOp | 0.534 | 0.233 | 0.170 | 0.363 | 0.349 | 0.013 | 0.067 |
| | CoCoOp | 0.505 | 0.352 | 0.252 | 0.253 | 0.010 | 0.060 | **0.000** |
| | VPT | 0.587 | 0.426 | 0.322 | 0.265 | 0.011 | 0.133 | **0.000** |
| | MaPLe | **0.599** | 0.430 | 0.311 | 0.288 | 0.008 | 0.137 | **0.000** |
| FGVCAircraft | ProGrad | 0.520 | 0.392 | 0.303 | 0.217 | 0.005 | 0.099 | **0.000** |
| | KgCoOp | 0.492 | 0.377 | 0.295 | 0.197 | 0.005 | 0.085 | **0.000** |
| | RPO | 0.536 | 0.410 | 0.322 | 0.214 | 0.006 | 0.118 | **0.000** |
| | PromptSRC | 0.564 | **0.437** | **0.352** | 0.213 | 0.008 | **0.145** | **0.000** |
| | ProDA | 0.495 | 0.387 | 0.310 | 0.185 | 0.005 | 0.095 | **0.000** |
| | TCP | 0.526 | 0.407 | 0.322 | 0.204 | 0.005 | 0.114 | **0.000** |

Table 15: Evaluation on Food101 under Emerging New Classes

| Dataset | Methods | Acc(0) | AUC | WA | EVM | VS | PA | NA |
|---------|---------|--------|-----|-----|-----|-----|-----|-----|
| | CLIP(Zero-shot) | 0.902 | 0.878 | 0.859 | **0.043** | **0.000** | / | / |
| | CoOp | 0.899 | 0.862 | 0.839 | 0.061 | 0.001 | 0.000 | 0.016 |
| | CoCoOp | 0.906 | 0.874 | 0.845 | 0.061 | **0.000** | 0.002 | 0.004 |
| | VPT | 0.896 | 0.856 | 0.820 | 0.076 | **0.000** | 0.000 | 0.022 |
| | MaPLe | 0.892 | 0.859 | 0.829 | 0.063 | **0.000** | 0.000 | 0.019 |
| Food101 | ProGrad | 0.912 | 0.884 | 0.858 | 0.054 | **0.000** | 0.006 | **0.000** |
| | KgCoOp | 0.914 | 0.888 | 0.865 | 0.049 | **0.000** | 0.010 | **0.000** |
| | RPO | 0.911 | 0.881 | 0.855 | 0.055 | **0.000** | 0.005 | 0.001 |
| | PromptSRC | 0.914 | 0.889 | 0.865 | 0.048 | **0.000** | 0.012 | **0.000** |
| | ProDA | **0.917** | **0.890** | **0.867** | 0.050 | **0.000** | **0.013** | **0.000** |
| | TCP | 0.913 | 0.888 | 0.865 | 0.048 | **0.000** | 0.010 | **0.000** |

Table 16: Evaluation on Flowers102 under Emerging New Classes

| Dataset | Methods | Acc(0) | AUC | WA | EVM | VS | PA | NA |
|---------|---------|--------|-----|-----|-----|-----|-----|-----|
| Flowers102 | CLIP(Zero-shot) | 0.736 | 0.729 | 0.707 | **0.049** | **0.002** | / | / |
| | CoOp | 0.975 | 0.707 | 0.656 | 0.325 | 0.299 | 0.020 | 0.040 |
| | CoCoOp | 0.969 | 0.826 | 0.729 | 0.240 | 0.009 | 0.097 | **0.000** |
| | VPT | 0.980 | 0.844 | 0.739 | 0.241 | 0.004 | 0.115 | **0.000** |
| | MaPLe | **0.981** | 0.855 | 0.770 | 0.211 | 0.006 | 0.125 | **0.000** |
| | ProGrad | 0.968 | 0.854 | 0.771 | 0.197 | 0.004 | 0.125 | **0.000** |
| | KgCoOp | 0.958 | 0.859 | 0.792 | 0.166 | 0.003 | 0.129 | **0.000** |
| | RPO | 0.977 | 0.860 | 0.788 | 0.189 | 0.006 | 0.131 | **0.000** |
| | PromptSRC | 0.978 | 0.881 | 0.809 | 0.169 | **0.002** | 0.152 | **0.000** |
| | ProDA | 0.974 | **0.882** | **0.814** | 0.160 | **0.002** | **0.153** | **0.000** |
| | TCP | 0.968 | 0.869 | 0.804 | 0.163 | 0.004 | 0.140 | **0.000** |

Table 17: Evaluation on OxfordPets under Emerging New Classes

| Dataset | Methods | Acc(0) | AUC | WA | EVM | VS | PA | NA |
|---------|---------|--------|-----|-----|-----|-----|-----|-----|
| OxfordPets | CLIP(Zero-shot) | 0.948 | 0.918 | 0.891 | 0.058 | **0.001** | / | / |
| | CoOp | 0.954 | 0.897 | 0.872 | 0.082 | 0.003 | 0.000 | 0.018 |
| | CoCoOp | 0.953 | 0.926 | 0.892 | 0.063 | 0.004 | 0.009 | 0.001 |
| | VPT | 0.956 | 0.932 | 0.904 | 0.054 | 0.003 | 0.013 | **0.000** |
| | MaPLe | 0.947 | 0.927 | 0.903 | 0.044 | 0.003 | 0.010 | 0.001 |
| | ProGrad | 0.957 | 0.935 | 0.904 | 0.056 | 0.003 | 0.017 | **0.000** |
| | KgCoOp | 0.953 | 0.940 | 0.916 | **0.040** | 0.004 | 0.022 | **0.000** |
| | RPO | 0.950 | 0.928 | 0.901 | 0.056 | 0.003 | 0.011 | 0.001 |
| | PromptSRC | 0.954 | 0.940 | 0.917 | 0.043 | 0.003 | 0.022 | **0.000** |
| | ProDA | **0.959** | **0.945** | **0.922** | 0.041 | 0.002 | **0.027** | **0.000** |
| | TCP | 0.954 | 0.939 | 0.916 | **0.040** | 0.003 | 0.021 | **0.000** |

Table 18: Evaluation on StanfordCars under Emerging New Classes

| Dataset | Methods | Acc(0) | AUC | WA | EVM | VS | PA | NA |
|---------|---------|--------|-----|-----|-----|-----|-----|-----|
| StanfordCars | CLIP(Zero-shot) | 0.757 | 0.706 | 0.655 | **0.102** | **0.001** | / | / |
| | CoOp | 0.858 | 0.685 | 0.618 | 0.240 | 0.066 | 0.008 | 0.027 |
| | CoCoOp | 0.845 | 0.753 | 0.682 | 0.163 | 0.002 | 0.046 | **0.000** |
| | VPT | 0.859 | 0.754 | 0.677 | 0.182 | 0.003 | 0.047 | **0.000** |
| | MaPLe | **0.894** | 0.775 | 0.688 | 0.206 | 0.004 | 0.069 | **0.000** |
| | ProGrad | 0.842 | 0.755 | 0.693 | 0.149 | 0.003 | 0.048 | **0.000** |
| | KgCoOp | 0.827 | 0.750 | 0.692 | 0.135 | 0.002 | 0.044 | **0.000** |
| | RPO | 0.851 | 0.772 | 0.712 | 0.139 | 0.002 | 0.065 | **0.000** |
| | PromptSRC | 0.866 | **0.783** | **0.722** | 0.144 | 0.002 | **0.077** | **0.000** |
| | ProDA | 0.822 | 0.755 | 0.704 | 0.118 | **0.001** | 0.048 | **0.000** |
| | TCP | 0.855 | 0.772 | 0.708 | 0.147 | 0.002 | 0.066 | **0.000** |

Table 19: Evaluation on SUN397 under Emerging New Classes

| Dataset | Methods | Acc(0) | AUC | WA | EVM | VS | PA | NA |
|---------|---------|--------|-----|-----|------|-----|-----|-----|
| SUN397 | CLIP(Zero-shot) | 0.732 | 0.672 | 0.626 | **0.107** | **0.001** | / | / |
| | CoOp | 0.835 | 0.677 | 0.621 | 0.214 | 0.064 | 0.010 | 0.004 |
| | CoCoOp | 0.830 | 0.740 | 0.670 | 0.160 | **0.001** | 0.069 | **0.000** |
| | VPT | 0.830 | 0.737 | 0.668 | 0.162 | 0.002 | 0.065 | **0.000** |
| | MaPLe | 0.833 | 0.740 | 0.669 | 0.163 | 0.002 | 0.068 | **0.000** |
| | ProGrad | 0.831 | 0.753 | 0.691 | 0.141 | **0.001** | 0.081 | **0.000** |
| | KgCoOp | 0.825 | 0.752 | 0.696 | 0.129 | **0.001** | 0.080 | **0.000** |
| | RPO | 0.845 | 0.761 | 0.701 | 0.145 | 0.002 | 0.090 | **0.000** |
| | PromptSRC | **0.848** | **0.776** | **0.722** | 0.126 | **0.001** | **0.105** | **0.000** |
| | ProDA | 0.837 | 0.770 | 0.717 | 0.120 | **0.001** | 0.098 | **0.000** |
| | TCP | 0.841 | 0.766 | 0.709 | 0.132 | **0.001** | 0.095 | **0.000** |

Table 20: Evaluation on UCF101 under Emerging New Classes

| Dataset | Methods | Acc(0) | AUC | WA | EVM | VS | PA | NA |
|---------|---------|--------|-----|-----|------|-----|-----|-----|
| UCF101 | CLIP(Zero-shot) | 0.748 | 0.708 | 0.675 | **0.073** | **0.000** | / | / |
| | CoOp | 0.895 | 0.696 | 0.641 | 0.254 | 0.137 | 0.012 | 0.023 |
| | CoCoOp | 0.880 | 0.760 | 0.683 | 0.196 | 0.009 | 0.052 | **0.000** |
| | VPT | 0.897 | 0.777 | 0.704 | 0.193 | 0.009 | 0.069 | **0.000** |
| | MaPLe | 0.896 | 0.782 | 0.713 | 0.183 | 0.008 | 0.075 | **0.000** |
| | ProGrad | 0.891 | 0.782 | 0.718 | 0.173 | 0.009 | 0.075 | **0.000** |
| | KgCoOp | 0.882 | 0.788 | 0.734 | 0.148 | 0.006 | 0.080 | **0.000** |
| | RPO | **0.909** | 0.792 | 0.730 | 0.179 | 0.012 | 0.084 | **0.000** |
| | PromptSRC | 0.904 | **0.812** | **0.755** | 0.149 | 0.005 | **0.104** | **0.000** |
| | ProDA | 0.884 | 0.782 | 0.725 | 0.160 | 0.008 | 0.074 | **0.000** |
| | TCP | 0.890 | 0.801 | 0.748 | 0.143 | 0.005 | 0.093 | **0.000** |

Table 21: Evaluation on ImageNet under Varying Ratio of New Classes

| Dataset | Methods | Acc(0) | AUC | WA | EVM | VS | PA | NA |
|---------|---------|--------|-----|-----|------|-----|-----|-----|
| ImageNet | CLIP(Zero-shot) | 0.754 | 0.754 | 0.748 | 0.018 | 0.001 | / | / |
| | CoOp | 0.796 | 0.773 | 0.765 | 0.043 | 0.003 | 0.019 | **0.000** |
| | CoCoOp | 0.795 | 0.783 | 0.770 | 0.026 | **0.000** | 0.028 | **0.000** |
| | VPT | 0.785 | 0.759 | 0.738 | 0.048 | **0.000** | 0.010 | 0.004 |
| | MaPLe | 0.792 | 0.753 | 0.717 | 0.075 | **0.000** | 0.010 | 0.010 |
| | ProGrad | 0.792 | 0.783 | 0.774 | 0.018 | **0.000** | 0.028 | **0.000** |
| | KgCoOp | 0.788 | 0.781 | 0.775 | 0.015 | **0.000** | 0.027 | **0.000** |
| | RPO | 0.801 | 0.789 | 0.778 | 0.023 | **0.000** | 0.035 | **0.000** |
| | PromptSRC | **0.807** | 0.794 | 0.782 | 0.025 | **0.000** | 0.040 | **0.000** |
| | ProDA | 0.800 | **0.795** | **0.791** | **0.009** | **0.000** | **0.041** | **0.000** |
| | TCP | 0.799 | 0.785 | 0.775 | 0.024 | **0.000** | 0.031 | **0.000** |

Table 22: Evaluation on Caltech101 under Varying Ratio of New Classes

| Dataset | Methods | Acc(0) | AUC | WA | EVM | VS | PA | NA |
|---------|---------|--------|-----|-----|-----|-----|-----|-----|
| | CLIP(Zero-shot) | 0.968 | 0.954 | 0.944 | 0.045 | 0.002 | / | / |
| | CoOp | 0.982 | 0.947 | 0.936 | 0.072 | 0.007 | 0.001 | 0.007 |
| | CoCoOp | 0.984 | 0.954 | 0.928 | 0.056 | 0.001 | 0.006 | 0.004 |
| | VPT | **0.988** | 0.962 | 0.939 | 0.049 | **0.000** | 0.011 | 0.003 |
| | MaPLe | 0.984 | 0.961 | 0.940 | 0.044 | **0.000** | 0.011 | 0.003 |
| Caltech101 | ProGrad | 0.986 | 0.964 | 0.933 | 0.053 | **0.000** | 0.013 | 0.003 |
| | KgCoOp | 0.983 | 0.968 | 0.947 | 0.036 | **0.000** | 0.015 | **0.001** |
| | RPO | 0.985 | **0.969** | **0.951** | 0.035 | **0.000** | **0.016** | **0.001** |
| | PromptSRC | 0.984 | 0.967 | 0.948 | 0.036 | **0.000** | 0.015 | **0.001** |
| | ProDA | 0.987 | 0.968 | 0.948 | 0.038 | **0.000** | **0.016** | **0.001** |
| | TCP | 0.985 | 0.968 | 0.950 | **0.034** | **0.000** | 0.015 | **0.001** |

Table 23: Evaluation on DTD under Varying Ratio of New Classes

| Dataset | Methods | Acc(0) | AUC | WA | EVM | VS | PA | NA |
|---------|---------|--------|-----|-----|-----|-----|-----|-----|
| | CLIP(Zero-shot) | 0.568 | 0.536 | 0.508 | **0.084** | **0.008** | / | / |
| | CoOp | 0.767 | 0.507 | 0.464 | 0.337 | 0.333 | 0.014 | 0.038 |
| | CoCoOp | 0.762 | 0.557 | 0.362 | 0.400 | 0.021 | 0.062 | 0.040 |
| | VPT | 0.799 | 0.606 | 0.471 | 0.328 | 0.022 | 0.081 | 0.007 |
| | MaPLe | 0.789 | 0.623 | 0.473 | 0.316 | 0.013 | 0.093 | 0.006 |
| DTD | ProGrad | 0.779 | 0.638 | 0.523 | 0.256 | 0.020 | 0.103 | **0.000** |
| | KgCoOp | 0.777 | 0.638 | 0.516 | 0.261 | 0.018 | 0.103 | 0.001 |
| | RPO | **0.812** | 0.656 | 0.538 | 0.273 | 0.016 | 0.120 | **0.000** |
| | PromptSRC | 0.807 | 0.663 | 0.552 | 0.255 | 0.015 | 0.127 | **0.000** |
| | ProDA | 0.779 | **0.670** | **0.575** | 0.204 | 0.013 | **0.135** | **0.000** |
| | TCP | 0.779 | 0.649 | 0.544 | 0.235 | 0.018 | 0.113 | **0.000** |

Table 24: Evaluation on EuroSAT under Varying Ratio of New Classes

| Dataset | Methods | Acc(0) | AUC | WA | EVM | VS | PA | NA |
|---------|---------|--------|-----|-----|-----|-----|-----|-----|
| | CLIP(Zero-shot) | 0.581 | 0.663 | 0.581 | 0.243 | 0.058 | / | / |
| | CoOp | 0.882 | 0.669 | 0.612 | 0.288 | 0.125 | 0.043 | 0.035 |
| | CoCoOp | 0.863 | 0.697 | 0.469 | 0.394 | 0.076 | 0.090 | 0.048 |
| | VPT | 0.950 | 0.805 | 0.672 | 0.412 | 0.126 | 0.143 | **0.000** |
| | MaPLe | **0.951** | 0.857 | 0.685 | 0.266 | 0.036 | 0.194 | **0.000** |
| EuroSAT | ProGrad | 0.887 | 0.730 | 0.554 | 0.333 | **0.021** | 0.097 | 0.027 |
| | KgCoOp | 0.854 | 0.744 | 0.593 | 0.262 | 0.037 | 0.092 | 0.009 |
| | RPO | 0.928 | 0.780 | 0.591 | 0.337 | 0.074 | 0.120 | 0.003 |
| | PromptSRC | 0.919 | **0.874** | **0.752** | **0.195** | 0.027 | **0.211** | **0.000** |
| | ProDA | 0.842 | 0.769 | 0.609 | 0.256 | 0.047 | 0.110 | 0.003 |
| | TCP | 0.882 | 0.777 | 0.647 | 0.271 | 0.053 | 0.115 | **0.000** |

Table 25: Evaluation on FGVCAircraft under Varying Ratio of New Classes

| Dataset | Methods | Acc(0) | AUC | WA | EVM | VS | PA | NA |
|---|---|---|---|---|---|---|---|---|
| | CLIP(Zero-shot) | 0.338 | 0.362 | 0.338 | **0.034** | **0.003** | / | / |
| | CoOp | 0.534 | 0.286 | 0.240 | 0.332 | 0.303 | 0.013 | 0.081 |
| | CoCoOp | 0.505 | 0.347 | 0.226 | 0.279 | 0.008 | 0.031 | 0.034 |
| | VPT | 0.587 | 0.459 | 0.376 | 0.210 | 0.012 | 0.097 | **0.000** |
| | MaPLe | **0.599** | 0.440 | 0.311 | 0.288 | 0.005 | 0.085 | 0.006 |
| FGVCAircraft | ProGrad | 0.520 | 0.423 | 0.350 | 0.170 | **0.003** | 0.062 | 0.001 |
| | KgCoOp | 0.492 | 0.413 | 0.356 | 0.136 | 0.005 | 0.051 | **0.000** |
| | RPO | 0.536 | 0.447 | 0.392 | 0.155 | 0.009 | 0.085 | **0.000** |
| | PromptSRC | 0.564 | **0.478** | **0.426** | 0.139 | 0.006 | **0.116** | **0.000** |
| | ProDA | 0.495 | 0.439 | 0.411 | 0.088 | 0.007 | 0.077 | **0.000** |
| | TCP | 0.526 | 0.445 | 0.394 | 0.132 | 0.006 | 0.084 | **0.000** |

Table 26: Evaluation on Food101 under Varying Ratio of New Classes

| Dataset | Methods | Acc(0) | AUC | WA | EVM | VS | PA | NA |
|---|---|---|---|---|---|---|---|---|
| | CLIP(Zero-shot) | 0.902 | 0.903 | 0.898 | 0.026 | 0.001 | / | / |
| | CoOp | 0.899 | 0.889 | 0.884 | 0.027 | 0.001 | 0.000 | 0.014 |
| | CoCoOp | 0.906 | 0.892 | 0.879 | 0.028 | 0.001 | 0.000 | 0.009 |
| | VPT | 0.896 | 0.872 | 0.849 | 0.047 | **0.000** | 0.000 | 0.031 |
| | MaPLe | 0.892 | 0.881 | 0.873 | 0.019 | **0.000** | 0.000 | 0.022 |
| Food101 | ProGrad | 0.912 | 0.901 | 0.892 | 0.020 | **0.000** | 0.004 | 0.005 |
| | KgCoOp | 0.914 | 0.907 | 0.902 | 0.015 | **0.000** | 0.005 | **0.000** |
| | RPO | 0.911 | 0.899 | 0.890 | 0.021 | **0.000** | 0.002 | 0.006 |
| | PromptSRC | 0.914 | 0.909 | 0.903 | 0.012 | **0.000** | 0.006 | 0.001 |
| | ProDA | **0.917** | **0.910** | **0.904** | 0.015 | **0.000** | **0.007** | **0.000** |
| | TCP | 0.913 | 0.908 | **0.904** | **0.010** | **0.000** | 0.006 | **0.000** |

Table 27: Evaluation on Flowers102 under Varying Ratio of New Classes

| Dataset | Methods | Acc(0) | AUC | WA | EVM | VS | PA | NA |
|---|---|---|---|---|---|---|---|---|
| | CLIP(Zero-shot) | 0.736 | 0.780 | 0.736 | **0.093** | 0.009 | / | / |
| | CoOp | 0.975 | 0.771 | 0.734 | 0.302 | 0.249 | 0.021 | 0.029 |
| | CoCoOp | 0.969 | 0.790 | 0.631 | 0.338 | 0.009 | 0.057 | 0.041 |
| | VPT | 0.980 | 0.810 | 0.649 | 0.331 | 0.004 | 0.066 | 0.034 |
| | MaPLe | **0.981** | 0.826 | 0.672 | 0.309 | 0.001 | 0.071 | 0.024 |
| Flowers102 | ProGrad | 0.968 | 0.829 | 0.696 | 0.272 | 0.004 | 0.069 | 0.019 |
| | KgCoOp | 0.958 | 0.842 | 0.741 | 0.217 | 0.003 | 0.071 | 0.006 |
| | RPO | 0.977 | 0.835 | 0.720 | 0.257 | 0.004 | 0.070 | 0.015 |
| | PromptSRC | 0.978 | **0.861** | 0.750 | 0.228 | 0.001 | **0.085** | **0.004** |
| | ProDA | 0.974 | **0.861** | 0.750 | 0.224 | **0.000** | **0.085** | **0.004** |
| | TCP | 0.968 | 0.849 | **0.751** | 0.217 | 0.003 | 0.075 | 0.005 |

Table 28: Evaluation on OxfordPets under Varying Ratio of New Classes

| Dataset | Methods | Acc(0) | AUC | WA | EVM | VS | PA | NA |
|---|---|---|---|---|---|---|---|---|
| | CLIP(Zero-shot) | 0.948 | 0.945 | 0.924 | 0.043 | **0.002** | / | / |
| | CoOp | 0.954 | 0.923 | 0.908 | 0.046 | **0.002** | 0.000 | 0.020 |
| | CoCoOp | 0.953 | 0.945 | 0.913 | 0.061 | 0.007 | 0.003 | 0.001 |
| | VPT | 0.956 | 0.953 | 0.931 | 0.045 | 0.005 | 0.008 | **0.000** |
| | MaPLe | 0.947 | 0.953 | 0.935 | 0.059 | 0.007 | 0.009 | **0.000** |
| OxfordPets | ProGrad | 0.957 | 0.951 | 0.922 | 0.054 | 0.005 | 0.007 | 0.001 |
| | KgCoOp | 0.953 | 0.962 | 0.945 | 0.058 | 0.005 | 0.017 | **0.000** |
| | RPO | 0.950 | 0.954 | 0.941 | 0.043 | 0.004 | 0.009 | 0.001 |
| | PromptSRC | 0.954 | 0.961 | 0.947 | 0.055 | 0.005 | 0.016 | **0.000** |
| | ProDA | **0.959** | **0.967** | **0.959** | **0.035** | **0.002** | **0.022** | **0.000** |
| | TCP | 0.954 | 0.963 | 0.951 | 0.048 | 0.003 | 0.018 | **0.000** |

Table 29: Evaluation on StanfordCars under Varying Ratio of New Classes

| Dataset | Methods | Acc(0) | AUC | WA | EVM | VS | PA | NA |
|---|---|---|---|---|---|---|---|---|
| | CLIP(Zero-shot) | 0.757 | 0.766 | 0.756 | 0.041 | 0.002 | / | / |
| | CoOp | 0.858 | 0.751 | 0.725 | 0.160 | 0.061 | 0.008 | 0.021 |
| | CoCoOp | 0.845 | 0.789 | 0.750 | 0.094 | 0.002 | 0.024 | 0.001 |
| | VPT | 0.859 | 0.781 | 0.731 | 0.128 | 0.003 | 0.023 | 0.006 |
| | MaPLe | **0.894** | 0.790 | 0.711 | 0.183 | 0.002 | 0.035 | 0.010 |
| StanfordCars | ProGrad | 0.842 | 0.794 | 0.769 | 0.073 | 0.003 | 0.028 | **0.000** |
| | KgCoOp | 0.827 | 0.795 | 0.779 | 0.048 | **0.001** | 0.029 | **0.000** |
| | RPO | 0.851 | 0.815 | 0.793 | 0.058 | **0.001** | 0.049 | **0.000** |
| | PromptSRC | 0.866 | **0.819** | 0.793 | 0.073 | 0.002 | **0.054** | **0.000** |
| | ProDA | 0.822 | 0.805 | **0.798** | **0.028** | **0.001** | 0.039 | **0.000** |
| | TCP | 0.855 | 0.809 | 0.778 | 0.077 | **0.001** | 0.044 | **0.000** |

Table 30: Evaluation on SUN397 under Varying Ratio of New Classes

| Dataset | Methods | Acc(0) | AUC | WA | EVM | VS | PA | NA |
|---|---|---|---|---|---|---|---|---|
| | CLIP(Zero-shot) | 0.732 | 0.725 | 0.723 | **0.019** | 0.001 | / | / |
| | CoOp | 0.835 | 0.728 | 0.714 | 0.133 | 0.059 | 0.010 | 0.006 |
| | CoCoOp | 0.830 | 0.768 | 0.700 | 0.130 | 0.001 | 0.046 | 0.002 |
| | VPT | 0.830 | 0.765 | 0.709 | 0.121 | **0.000** | 0.041 | 0.001 |
| | MaPLe | 0.833 | 0.765 | 0.698 | 0.135 | **0.000** | 0.043 | 0.003 |
| SUN397 | ProGrad | 0.831 | 0.783 | 0.732 | 0.099 | **0.000** | 0.058 | **0.000** |
| | KgCoOp | 0.825 | 0.788 | 0.754 | 0.071 | **0.000** | 0.063 | **0.000** |
| | RPO | 0.845 | 0.790 | 0.745 | 0.100 | 0.001 | 0.065 | **0.000** |
| | PromptSRC | **0.848** | **0.808** | 0.772 | 0.075 | **0.000** | **0.084** | **0.000** |
| | ProDA | 0.837 | 0.805 | **0.776** | 0.061 | **0.000** | 0.080 | **0.000** |
| | TCP | 0.841 | 0.798 | 0.760 | 0.081 | **0.000** | 0.073 | **0.000** |

Table 31: Evaluation on UCF101 under Varying Ratio of New Classes

| Dataset | Methods | Acc(0) | AUC | WA | EVM | VS | PA | NA |
|---------|---------|--------|-----|-----|-----|-----|-----|-----|
| UCF101 | CLIP(Zero-shot) | 0.748 | 0.756 | 0.730 | **0.125** | 0.021 | / | / |
| | CoOp | 0.895 | 0.749 | 0.700 | 0.288 | 0.181 | 0.012 | 0.019 |
| | CoCoOp | 0.880 | 0.766 | 0.669 | 0.211 | **0.004** | 0.039 | 0.029 |
| | VPT | 0.897 | 0.777 | 0.675 | 0.222 | 0.005 | 0.046 | 0.010 |
| | MaPLe | 0.896 | 0.786 | 0.697 | 0.199 | **0.004** | 0.049 | 0.008 |
| | ProGrad | 0.891 | 0.789 | 0.704 | 0.187 | **0.004** | 0.050 | 0.008 |
| | KgCoOp | 0.882 | 0.804 | 0.739 | 0.143 | **0.004** | 0.054 | 0.005 |
| | RPO | **0.909** | 0.793 | 0.710 | 0.199 | 0.009 | 0.053 | 0.008 |
| | PromptSRC | 0.904 | **0.823** | 0.753 | 0.151 | **0.004** | **0.070** | 0.003 |
| | ProDA | 0.884 | 0.801 | 0.736 | 0.148 | 0.005 | 0.053 | 0.005 |
| | TCP | 0.890 | 0.821 | **0.761** | 0.129 | **0.004** | 0.066 | **0.002** |

Table 32: Evaluation on ImageNet-A under Dynamic Distribution shifts

| Dataset | Methods | Acc(0) | AUC | WA | EVM | VS | PA | NA |
|---------|---------|--------|-----|-----|-----|-----|-----|-----|
| ImageNet-A | CLIP(Zero-shot) | 0.904 | 0.708 | 0.477 | 0.427 | 0.004 | / | / |
| | CoOp | 0.925 | 0.733 | 0.505 | 0.420 | **0.004** | 0.025 | **0.000** |
| | CoCoOp | 0.923 | 0.734 | 0.509 | 0.415 | **0.004** | 0.026 | **0.000** |
| | VPT | 0.911 | 0.710 | 0.475 | 0.436 | **0.004** | 0.002 | 0.001 |
| | MaPLe | 0.924 | 0.725 | 0.490 | 0.435 | **0.004** | 0.017 | **0.000** |
| | ProGrad | 0.920 | 0.729 | 0.504 | 0.416 | **0.004** | 0.020 | **0.000** |
| | KgCoOp | 0.924 | 0.733 | 0.507 | 0.417 | **0.004** | 0.025 | **0.000** |
| | RPO | 0.922 | 0.730 | 0.502 | 0.420 | **0.004** | 0.021 | **0.000** |
| | PromptSRC | 0.922 | 0.732 | 0.508 | 0.414 | **0.004** | 0.024 | **0.000** |
| | ProDA | **0.926** | **0.740** | **0.520** | **0.406** | **0.004** | **0.032** | **0.000** |
| | TCP | 0.921 | 0.730 | 0.506 | 0.415 | 0.004 | 0.022 | **0.000** |

Table 33: Evaluation on ImageNet-R under Dynamic Distribution shifts

| Dataset | Methods | Acc(0) | AUC | WA | EVM | VS | PA | NA |
|---------|---------|--------|-----|-----|-----|-----|-----|-----|
| ImageNet-R | CLIP(Zero-shot) | 0.915 | 0.797 | 0.740 | 0.175 | 0.014 | / | / |
| | CoOp | 0.927 | 0.815 | 0.761 | 0.166 | 0.013 | 0.017 | **0.000** |
| | CoCoOp | 0.928 | 0.818 | 0.764 | 0.164 | 0.013 | 0.020 | **0.000** |
| | VPT | 0.920 | 0.809 | 0.755 | 0.166 | 0.012 | 0.012 | **0.000** |
| | MaPLe | **0.930** | 0.816 | 0.762 | 0.168 | 0.013 | 0.019 | **0.000** |
| | ProGrad | 0.925 | 0.818 | 0.766 | 0.159 | 0.011 | 0.021 | **0.000** |
| | KgCoOp | 0.929 | 0.821 | 0.768 | 0.160 | 0.012 | 0.024 | **0.000** |
| | RPO | 0.928 | 0.818 | 0.763 | 0.165 | 0.012 | 0.020 | **0.000** |
| | PromptSRC | 0.928 | 0.826 | 0.776 | 0.152 | **0.010** | 0.029 | **0.000** |
| | ProDA | 0.929 | **0.827** | **0.779** | **0.150** | 0.011 | **0.030** | **0.000** |
| | TCP | 0.927 | 0.819 | 0.766 | 0.160 | 0.011 | 0.022 | **0.000** |

Table 34: Evaluation on ImageNet-Sketch under Dynamic Distribution shifts

| Dataset | Methods | Acc(0) | AUC | WA | EVM | VS | PA | NA |
|---|---|---|---|---|---|---|---|---|
| | CLIP(Zero-shot) | 0.669 | 0.563 | 0.463 | **0.206** | **0.000** | / | / |
| | CoOp | 0.717 | 0.600 | 0.489 | 0.228 | **0.000** | 0.038 | **0.000** |
| | CoCoOp | 0.711 | 0.597 | 0.490 | 0.221 | **0.000** | 0.034 | **0.000** |
| | VPT | 0.685 | 0.578 | 0.475 | 0.211 | **0.000** | 0.015 | **0.000** |
| | MaPLe | **0.722** | **0.601** | 0.486 | 0.236 | **0.000** | **0.039** | **0.000** |
| ImageNet-Sketch | ProGrad | 0.704 | 0.594 | 0.488 | 0.215 | **0.000** | 0.031 | **0.000** |
| | KgCoOp | 0.712 | **0.601** | 0.495 | 0.217 | **0.000** | 0.038 | **0.000** |
| | RPO | 0.708 | 0.596 | 0.488 | 0.220 | **0.000** | 0.033 | **0.000** |
| | PromptSRC | 0.711 | **0.601** | 0.496 | 0.216 | **0.000** | 0.038 | **0.000** |
| | ProDA | 0.712 | **0.601** | **0.497** | 0.215 | **0.000** | 0.038 | **0.000** |
| | TCP | 0.711 | 0.597 | 0.489 | 0.222 | **0.000** | 0.034 | **0.000** |

Table 35: Evaluation on ImageNetV2 under Dynamic Distribution shifts

| Dataset | Methods | Acc(0) | AUC | WA | EVM | VS | PA | NA |
|---|---|---|---|---|---|---|---|---|
| | CLIP(Zero-shot) | 0.669 | 0.651 | 0.609 | **0.060** | **0.002** | / | / |
| | CoOp | 0.717 | 0.695 | 0.644 | 0.074 | 0.004 | 0.045 | **0.000** |
| | CoCoOp | 0.711 | 0.691 | 0.643 | 0.068 | 0.003 | 0.040 | **0.000** |
| | VPT | 0.685 | 0.665 | 0.616 | 0.070 | 0.003 | 0.014 | **0.000** |
| | MaPLe | **0.722** | **0.701** | **0.649** | 0.073 | 0.004 | **0.050** | **0.000** |
| ImageNetV2 | ProGrad | 0.704 | 0.683 | 0.633 | 0.071 | 0.004 | 0.032 | **0.000** |
| | KgCoOp | 0.712 | 0.692 | 0.644 | 0.068 | 0.003 | 0.041 | **0.000** |
| | RPO | 0.708 | 0.688 | 0.640 | 0.068 | 0.003 | 0.037 | **0.000** |
| | PromptSRC | 0.711 | 0.691 | 0.644 | 0.067 | 0.003 | 0.041 | **0.000** |
| | ProDA | 0.712 | 0.692 | 0.646 | 0.066 | 0.003 | 0.041 | **0.000** |
| | TCP | 0.711 | 0.690 | 0.640 | 0.071 | 0.004 | 0.040 | **0.000** |

Table 36: Robustness and Ranks of Prompt Learning Methods on ImageNet under Emerging New Classes

| Dataset | Methods | $\delta_{AUC}$ | $\delta_{PN}$ | Friedman rank | Final rank |
|---|---|---|---|---|---|
| | CLIP(Zero-shot) | / | / | 10.500 | 11 |
| | CoOp | 0.019 | 0.019 | 8.167 | 8 |
| | CoCoOp | 0.035 | 0.035 | 5.667 | 5 |
| | VPT | 0.013 | 0.013 | 9.333 | 10 |
| | MaPLe | 0.013 | 0.013 | 9.167 | 9 |
| ImageNet | ProGrad | 0.034 | 0.034 | 6.000 | 6 |
| | KgCoOp | 0.031 | 0.031 | 7.167 | 7 |
| | RPO | 0.041 | 0.041 | 2.833 | 3 |
| | PromptSRC | **0.047** | **0.047** | 1.500 | 1 |
| | ProDA | 0.045 | 0.045 | 1.667 | 2 |
| | TCP | 0.039 | 0.039 | 4.000 | 4 |

Table 37: Robustness and Ranks of Prompt Learning Methods on Caltech101 under Emerging New Classes

| Dataset | Methods | $\delta_{AUC}$ | $\delta_{PN}$ | Friedman rank | Final rank |
|---|---|---|---|---|---|
| | CLIP(Zero-shot) | / | / | 10.000 | 10 |
| | CoOp | -0.012 | -0.010 | 10.833 | 11 |
| | CoCoOp | 0.005 | 0.005 | 8.833 | 9 |
| | VPT | 0.011 | 0.011 | 6.000 | 7 |
| | MaPLe | 0.010 | 0.010 | 7.500 | 8 |
| Caltech101 | ProGrad | 0.017 | 0.017 | 3.333 | 2 |
| | KgCoOp | 0.016 | 0.016 | 4.500 | 5 |
| | RPO | 0.015 | 0.015 | 5.000 | 6 |
| | PromptSRC | 0.016 | 0.016 | 4.333 | 4 |
| | ProDA | **0.019** | **0.019** | 1.500 | 1 |
| | TCP | 0.016 | 0.016 | 4.167 | 3 |

Table 38: Robustness and Ranks of Prompt Learning Methods on DTD under Emerging New Classes

| Dataset | Methods | $\delta_{AUC}$ | $\delta_{PN}$ | Friedman rank | Final rank |
|---|---|---|---|---|---|
| | CLIP(Zero-shot) | / | / | 10.167 | 10 |
| | CoOp | -0.020 | -0.017 | 10.667 | 11 |
| | CoCoOp | 0.108 | 0.108 | 9.167 | 9 |
| | VPT | 0.129 | 0.129 | 7.167 | 8 |
| | MaPLe | 0.144 | 0.144 | 5.833 | 6 |
| DTD | ProGrad | 0.140 | 0.140 | 6.500 | 7 |
| | KgCoOp | 0.147 | 0.147 | 5.667 | 5 |
| | RPO | 0.166 | 0.166 | 2.500 | 3 |
| | PromptSRC | **0.172** | **0.172** | 1.833 | 1 |
| | ProDA | 0.168 | 0.168 | 2.167 | 2 |
| | TCP | 0.152 | 0.152 | 4.333 | 4 |

Table 39: Robustness and Ranks of Prompt Learning Methods on EuroSAT under Emerging New Classes

| Dataset | Methods | $\delta_{AUC}$ | $\delta_{PN}$ | Friedman rank | Final rank |
|---|---|---|---|---|---|
| | CLIP(Zero-shot) | / | / | 10.167 | 11 |
| | CoOp | 0.012 | 0.014 | 10.000 | 10 |
| | CoCoOp | 0.106 | 0.106 | 9.000 | 9 |
| | VPT | 0.204 | 0.204 | 3.167 | 3 |
| | MaPLe | 0.265 | 0.265 | 1.667 | 1 |
| EuroSAT | ProGrad | 0.154 | 0.154 | 6.833 | 7 |
| | KgCoOp | 0.149 | 0.149 | 7.667 | 8 |
| | RPO | 0.194 | 0.194 | 3.500 | 4 |
| | PromptSRC | **0.275** | **0.275** | 1.667 | 2 |
| | ProDA | 0.155 | 0.155 | 6.833 | 6 |
| | TCP | 0.166 | 0.166 | 5.500 | 5 |

Table 40: Robustness and Ranks of Prompt Learning Methods on FGVCAircraft under Emerging New Classes

| Dataset | Methods | $\delta_{AUC}$ | $\delta_{PN}$ | Friedman rank | Final rank |
|---|---|---|---|---|---|
| | CLIP(Zero-shot) | / | / | 10.167 | 11 |
| | CoOp | -0.060 | -0.054 | 10.000 | 10 |
| | CoCoOp | 0.060 | 0.060 | 8.833 | 9 |
| | VPT | 0.133 | 0.133 | 2.667 | 2 |
| | MaPLe | 0.137 | 0.137 | 2.667 | 3 |
| FGVCAircraft | ProGrad | 0.099 | 0.099 | 6.667 | 6 |
| | KgCoOp | 0.085 | 0.085 | 8.333 | 8 |
| | RPO | 0.118 | 0.118 | 3.667 | 4 |
| | PromptSRC | **0.145** | **0.145** | 1.667 | 1 |
| | ProDA | 0.095 | 0.095 | 6.833 | 7 |
| | TCP | 0.114 | 0.114 | 4.500 | 5 |

Table 41: Robustness and Ranks of Prompt Learning Methods on Food101 under Emerging New Classes

| Dataset | Methods | $\delta_{AUC}$ | $\delta_{PN}$ | Friedman rank | Final rank |
|---|---|---|---|---|---|
| | CLIP(Zero-shot) | / | / | 6.667 | 7 |
| | CoOp | -0.016 | -0.016 | 9.667 | 9 |
| | CoCoOp | -0.004 | -0.003 | 7.500 | 8 |
| | VPT | -0.022 | -0.022 | 10.500 | 11 |
| | MaPLe | -0.019 | -0.019 | 9.833 | 10 |
| Food101 | ProGrad | 0.006 | 0.006 | 5.333 | 5 |
| | KgCoOp | 0.010 | 0.010 | 3.500 | 4 |
| | RPO | 0.003 | 0.003 | 6.500 | 6 |
| | PromptSRC | 0.012 | 0.012 | 1.833 | 2 |
| | ProDA | **0.013** | **0.013** | 1.333 | 1 |
| | TCP | 0.010 | 0.010 | 3.333 | 3 |

Table 42: Robustness and Ranks of Prompt Learning Methods on Flowers102 under Emerging New Classes

| Dataset | Methods | $\delta_{AUC}$ | $\delta_{PN}$ | Friedman rank | Final rank |
|---|---|---|---|---|---|
| | CLIP(Zero-shot) | / | / | 10.167 | 11 |
| | CoOp | -0.022 | -0.020 | 10.000 | 10 |
| | CoCoOp | 0.097 | 0.097 | 8.667 | 9 |
| | VPT | 0.115 | 0.115 | 6.333 | 8 |
| | MaPLe | 0.125 | 0.125 | 5.500 | 5 |
| Flowers102 | ProGrad | 0.125 | 0.125 | 6.167 | 7 |
| | KgCoOp | 0.129 | 0.129 | 6.167 | 6 |
| | RPO | 0.131 | 0.131 | 4.833 | 4 |
| | PromptSRC | 0.152 | 0.152 | 1.833 | 1 |
| | ProDA | **0.153** | **0.153** | 2.167 | 2 |
| | TCP | 0.140 | 0.140 | 4.167 | 3 |

Table 43: Robustness and Ranks of Prompt Learning Methods on OxfordPets under Emerging New Classes

| Dataset | Methods | $\delta_{AUC}$ | $\delta_{PN}$ | Friedman rank | Final rank |
|---|---|---|---|---|---|
| | CLIP(Zero-shot) | / | / | 9.333 | 10 |
| | CoOp | -0.021 | -0.018 | 10.167 | 11 |
| | CoCoOp | 0.008 | 0.008 | 8.500 | 8 |
| | VPT | 0.013 | 0.013 | 5.500 | 6 |
| | MaPLe | 0.009 | 0.010 | 8.667 | 9 |
| OxfordPets | ProGrad | 0.017 | 0.017 | 4.167 | 4 |
| | KgCoOp | 0.022 | 0.022 | 3.333 | 2 |
| | RPO | 0.010 | 0.010 | 7.667 | 7 |
| | PromptSRC | 0.022 | 0.022 | 3.333 | 3 |
| | ProDA | **0.027** | **0.027** | 1.000 | 1 |
| | TCP | 0.021 | 0.021 | 4.333 | 5 |

Table 44: Robustness and Ranks of Prompt Learning Methods on StanfordCars under Emerging New Classes

| Dataset | Methods | $\delta_{AUC}$ | $\delta_{PN}$ | Friedman rank | Final rank |
|---|---|---|---|---|---|
| | CLIP(Zero-shot) | / | / | 10.167 | 11 |
| | CoOp | -0.021 | -0.019 | 9.833 | 10 |
| | CoCoOp | 0.046 | 0.046 | 7.500 | 8 |
| | VPT | 0.047 | 0.047 | 6.667 | 7 |
| | MaPLe | 0.069 | 0.069 | 3.333 | 4 |
| StanfordCars | ProGrad | 0.048 | 0.048 | 6.500 | 6 |
| | KgCoOp | 0.044 | 0.044 | 7.667 | 9 |
| | RPO | 0.065 | 0.065 | 3.333 | 2 |
| | PromptSRC | **0.077** | **0.077** | 1.333 | 1 |
| | ProDA | 0.048 | 0.048 | 6.333 | 5 |
| | TCP | 0.066 | 0.066 | 3.333 | 3 |

Table 45: Robustness and Ranks of Prompt Learning Methods on SUN397 under Emerging New Classes

| Dataset | Methods | $\delta_{AUC}$ | $\delta_{PN}$ | Friedman rank | Final rank |
|---|---|---|---|---|---|
| | CLIP(Zero-shot) | / | / | 10.167 | 11 |
| | CoOp | 0.005 | 0.005 | 10.000 | 10 |
| | CoCoOp | 0.069 | 0.069 | 7.500 | 8 |
| | VPT | 0.065 | 0.065 | 8.833 | 9 |
| | MaPLe | 0.068 | 0.068 | 7.500 | 7 |
| SUN397 | ProGrad | 0.081 | 0.081 | 5.667 | 5 |
| | KgCoOp | 0.080 | 0.080 | 6.333 | 6 |
| | RPO | 0.090 | 0.090 | 3.667 | 4 |
| | PromptSRC | **0.105** | **0.105** | 1.000 | 1 |
| | ProDA | 0.098 | 0.098 | 2.333 | 2 |
| | TCP | 0.095 | 0.095 | 3.000 | 3 |

Table 46: Robustness and Ranks of Prompt Learning Methods on UCF101 under Emerging New Classes

| Dataset | Methods | $\delta_{AUC}$ | $\delta_{PN}$ | Friedman rank | Final rank |
|---------|---------|----------|---------|---------------|------------|
| | CLIP(Zero-shot) | / | / | 10.167 | 11 |
| | CoOp | -0.012 | -0.011 | 10.000 | 10 |
| | CoCoOp | 0.052 | 0.052 | 9.167 | 9 |
| | VPT | 0.069 | 0.069 | 6.833 | 8 |
| | MaPLe | 0.075 | 0.075 | 5.833 | 5 |
| UCF101 | ProGrad | 0.075 | 0.075 | 6.000 | 6 |
| | KgCoOp | 0.080 | 0.080 | 4.667 | 4 |
| | RPO | 0.084 | 0.084 | 3.000 | 3 |
| | PromptSRC | **0.104** | **0.104** | 1.167 | 1 |
| | ProDA | 0.074 | 0.074 | 6.333 | 7 |
| | TCP | 0.093 | 0.093 | 2.833 | 2 |

Table 47: Robustness and Ranks of Prompt Learning Methods on ImageNet under Varying Ratio of New Classes

| Dataset | Methods | $\delta_{AUC}$ | $\delta_{PN}$ | Friedman rank | Final rank |
|---------|---------|----------|---------|---------------|------------|
| | CLIP(Zero-shot) | / | / | 10.000 | 11 |
| | CoOp | 0.019 | 0.019 | 7.667 | 8 |
| | CoCoOp | 0.028 | 0.028 | 6.167 | 6 |
| | VPT | 0.005 | 0.005 | 9.667 | 9 |
| | MaPLe | -0.001 | 0.000 | 9.667 | 10 |
| ImageNet | ProGrad | 0.028 | 0.028 | 6.000 | 5 |
| | KgCoOp | 0.027 | 0.027 | 6.667 | 7 |
| | RPO | 0.035 | 0.035 | 2.833 | 3 |
| | PromptSRC | 0.040 | 0.040 | 1.500 | 1 |
| | ProDA | **0.041** | **0.041** | 1.667 | 2 |
| | TCP | 0.031 | 0.031 | 4.167 | 4 |

Table 48: Robustness and Ranks of Prompt Learning Methods on Caltech101 under Varying Ratio of New Classes

| Dataset | Methods | $\delta_{AUC}$ | $\delta_{PN}$ | Friedman rank | Final rank |
|---------|---------|----------|---------|---------------|------------|
| | CLIP(Zero-shot) | / | / | 8.000 | 9 |
| | CoOp | -0.007 | -0.006 | 9.333 | 11 |
| | CoCoOp | 0.000 | 0.002 | 9.167 | 10 |
| | VPT | 0.008 | 0.009 | 6.833 | 7 |
| | MaPLe | 0.008 | 0.008 | 7.833 | 8 |
| Caltech101 | ProGrad | 0.010 | 0.010 | 5.500 | 6 |
| | KgCoOp | 0.014 | 0.014 | 4.667 | 4 |
| | RPO | **0.015** | **0.015** | 2.667 | 1 |
| | PromptSRC | 0.013 | 0.013 | 4.667 | 5 |
| | ProDA | 0.014 | 0.014 | 3.667 | 2 |
| | TCP | 0.014 | 0.014 | 3.667 | 3 |

Table 49: Robustness and Ranks of Prompt Learning Methods on DTD under Varying Ratio of New Classes

| Dataset | Methods | $\delta_{AUC}$ | $\delta_{PN}$ | Friedman rank | Final rank |
|---|---|---|---|---|---|
| | CLIP(Zero-shot) | / | / | 8.667 | 9 |
| | CoOp | -0.029 | -0.024 | 10.000 | 11 |
| | CoCoOp | 0.021 | 0.022 | 10.000 | 10 |
| | VPT | 0.070 | 0.073 | 7.667 | 8 |
| | MaPLe | 0.087 | 0.087 | 6.833 | 7 |
| DTD | ProGrad | 0.102 | 0.102 | 5.667 | 5 |
| | KgCoOp | 0.103 | 0.103 | 6.000 | 6 |
| | RPO | 0.120 | 0.120 | 2.667 | 3 |
| | PromptSRC | 0.127 | 0.127 | 2.000 | 1 |
| | ProDA | **0.135** | **0.135** | 2.167 | 2 |
| | TCP | 0.113 | 0.113 | 4.333 | 4 |

Table 50: Robustness and Ranks of Prompt Learning Methods on EuroSAT under Varying Ratio of New Classes

| Dataset | Methods | $\delta_{AUC}$ | $\delta_{PN}$ | Friedman rank | Final rank |
|---|---|---|---|---|---|
| | CLIP(Zero-shot) | / | / | 8.167 | 8 |
| | CoOp | 0.007 | 0.008 | 8.667 | 10 |
| | CoCoOp | 0.035 | 0.042 | 9.333 | 11 |
| | VPT | 0.143 | 0.143 | 3.500 | 3 |
| | MaPLe | 0.194 | 0.194 | 1.833 | 2 |
| EuroSAT | ProGrad | 0.068 | 0.070 | 8.167 | 9 |
| | KgCoOp | 0.081 | 0.083 | 8.000 | 7 |
| | RPO | 0.117 | 0.117 | 5.167 | 4 |
| | PromptSRC | **0.211** | **0.211** | 1.667 | 1 |
| | ProDA | 0.107 | 0.107 | 6.000 | 6 |
| | TCP | 0.115 | 0.115 | 5.500 | 5 |

Table 51: Robustness and Ranks of Prompt Learning Methods on FGVCAircraft under Varying Ratio of New Classes

| Dataset | Methods | $\delta_{AUC}$ | $\delta_{PN}$ | Friedman rank | Final rank |
|---|---|---|---|---|---|
| | CLIP(Zero-shot) | / | / | 9.000 | 9 |
| | CoOp | -0.076 | -0.068 | 9.833 | 11 |
| | CoCoOp | -0.015 | -0.003 | 9.500 | 10 |
| | VPT | 0.097 | 0.097 | 3.500 | 2 |
| | MaPLe | 0.078 | 0.079 | 5.000 | 6 |
| FGVCAircraft | ProGrad | 0.061 | 0.061 | 6.667 | 7 |
| | KgCoOp | 0.051 | 0.051 | 7.833 | 8 |
| | RPO | 0.085 | 0.085 | 4.000 | 3 |
| | PromptSRC | **0.116** | **0.116** | 1.667 | 1 |
| | ProDA | 0.077 | 0.077 | 4.667 | 5 |
| | TCP | 0.084 | 0.084 | 4.333 | 4 |

Table 52: Robustness and Ranks of Prompt Learning Methods on Food101 under Varying Ratio of New Classes

| Dataset | Methods | $\delta_{AUC}$ | $\delta_{PN}$ | Friedman rank | Final rank |
|---------|---------|----------------|---------------|---------------|------------|
| Food101 | CLIP(Zero-shot) | / | / | 5.000 | 5 |
| | CoOp | -0.014 | -0.014 | 8.333 | 9 |
| | CoCoOp | -0.012 | -0.008 | 8.167 | 8 |
| | VPT | -0.031 | -0.031 | 10.833 | 11 |
| | MaPLe | -0.022 | -0.022 | 10.167 | 10 |
| | ProGrad | -0.002 | -0.002 | 5.500 | 6 |
| | KgCoOp | 0.004 | 0.005 | 3.667 | 4 |
| | RPO | -0.005 | -0.003 | 6.667 | 7 |
| | PromptSRC | 0.005 | 0.006 | 2.667 | 2 |
| | ProDA | **0.006** | **0.007** | 1.500 | 1 |
| | TCP | 0.004 | 0.005 | 3.500 | 3 |

Table 53: Robustness and Ranks of Prompt Learning Methods on Flowers102 under Varying Ratio of New Classes

| Dataset | Methods | $\delta_{AUC}$ | $\delta_{PN}$ | Friedman rank | Final rank |
|---------|---------|----------------|---------------|---------------|------------|
| Flowers102 | CLIP(Zero-shot) | / | / | 6.667 | 7 |
| | CoOp | -0.009 | -0.008 | 7.500 | 9 |
| | CoCoOp | 0.010 | 0.016 | 9.667 | 11 |
| | VPT | 0.030 | 0.032 | 8.000 | 10 |
| | MaPLe | 0.046 | 0.046 | 6.000 | 6 |
| | ProGrad | 0.049 | 0.050 | 7.167 | 8 |
| | KgCoOp | 0.062 | 0.065 | 6.000 | 5 |
| | RPO | 0.055 | 0.055 | 5.667 | 4 |
| | PromptSRC | 0.081 | 0.081 | 2.333 | 1 |
| | ProDA | **0.082** | **0.082** | 2.667 | 2 |
| | TCP | 0.069 | 0.070 | 4.333 | 3 |

Table 54: Robustness and Ranks of Prompt Learning Methods on OxfordPets under Varying Ratio of New Classes

| Dataset | Methods | $\delta_{AUC}$ | $\delta_{PN}$ | Friedman rank | Final rank |
|---------|---------|----------------|---------------|---------------|------------|
| OxfordPets | CLIP(Zero-shot) | / | / | 8.833 | 9 |
| | CoOp | -0.022 | -0.020 | 10.167 | 11 |
| | CoCoOp | 0.000 | 0.002 | 9.000 | 10 |
| | VPT | 0.008 | 0.008 | 6.333 | 6 |
| | MaPLe | 0.009 | 0.009 | 7.000 | 8 |
| | ProGrad | 0.006 | 0.006 | 6.333 | 5 |
| | KgCoOp | 0.017 | 0.017 | 3.667 | 4 |
| | RPO | 0.009 | 0.009 | 6.833 | 7 |
| | PromptSRC | 0.016 | 0.016 | 3.333 | 3 |
| | ProDA | **0.022** | **0.022** | 1.333 | 1 |
| | TCP | 0.018 | 0.018 | 3.167 | 2 |

Table 55: Robustness and Ranks of Prompt Learning Methods on StanfordCars under Varying Ratio of New Classes

| Dataset | Methods | $\delta_{AUC}$ | $\delta_{PN}$ | Friedman rank | Final rank |
|---|---|---|---|---|---|
| | CLIP(Zero-shot) | / | / | 8.833 | 10 |
| | CoOp | -0.015 | -0.014 | 9.667 | 11 |
| | CoCoOp | 0.023 | 0.023 | 7.167 | 8 |
| | VPT | 0.016 | 0.017 | 7.667 | 9 |
| | MaPLe | 0.025 | 0.025 | 6.167 | 6 |
| StanfordCars | ProGrad | 0.028 | 0.028 | 6.667 | 7 |
| | KgCoOp | 0.029 | 0.029 | 6.167 | 5 |
| | RPO | 0.049 | 0.049 | 3.333 | 2 |
| | PromptSRC | **0.054** | **0.054** | 1.500 | 1 |
| | ProDA | 0.039 | 0.039 | 4.833 | 4 |
| | TCP | 0.044 | 0.044 | 4.000 | 3 |

Table 56: Robustness and Ranks of Prompt Learning Methods on SUN397 under Varying Ratio of New Classes

| Dataset | Methods | $\delta_{AUC}$ | $\delta_{PN}$ | Friedman rank | Final rank |
|---|---|---|---|---|---|
| | CLIP(Zero-shot) | / | / | 9.500 | 10 |
| | CoOp | 0.003 | 0.003 | 9.500 | 11 |
| | CoCoOp | 0.043 | 0.043 | 8.000 | 7 |
| | VPT | 0.040 | 0.040 | 8.333 | 8 |
| | MaPLe | 0.040 | 0.040 | 8.667 | 9 |
| SUN397 | ProGrad | 0.058 | 0.058 | 6.000 | 6 |
| | KgCoOp | 0.063 | 0.063 | 5.500 | 5 |
| | RPO | 0.065 | 0.065 | 4.167 | 4 |
| | PromptSRC | **0.084** | **0.084** | 1.167 | 1 |
| | ProDA | 0.080 | 0.080 | 2.167 | 2 |
| | TCP | 0.073 | 0.073 | 3.000 | 3 |

Table 57: Robustness and Ranks of Prompt Learning Methods on UCF101 under Varying Ratio of New Classes

| Dataset | Methods | $\delta_{AUC}$ | $\delta_{PN}$ | Friedman rank | Final rank |
|---|---|---|---|---|---|
| | CLIP(Zero-shot) | / | / | 7.000 | 8 |
| | CoOp | -0.007 | -0.006 | 8.000 | 10 |
| | CoCoOp | 0.010 | 0.010 | 10.000 | 11 |
| | VPT | 0.021 | 0.036 | 7.833 | 9 |
| | MaPLe | 0.030 | 0.041 | 6.500 | 6 |
| UCF101 | ProGrad | 0.033 | 0.042 | 6.500 | 7 |
| | KgCoOp | 0.047 | 0.050 | 5.167 | 4 |
| | RPO | 0.037 | 0.045 | 5.000 | 3 |
| | PromptSRC | **0.066** | **0.067** | 1.833 | 1 |
| | ProDA | 0.045 | 0.048 | 5.333 | 5 |
| | TCP | 0.065 | 0.065 | 2.833 | 2 |

