# OpenReview forum: "OpenPL: Realistic Evaluation of Prompt Learning for VLM in Open Environments"
_ICLR.cc/2025/Conference — Submitted to ICLR 2025_

### Official Review · Reviewer_owVA · 2024-11-01

**Soundness:** 3
**Presentation:** 3
**Contribution:** 3
**Rating:** 6
**Confidence:** 4

**Summary:**

This work explores the application of Vision-Language Models (VLMs) in practical open environments, where data distributions and classes are often uncertain and continuously evolving. To better assess the capability of current prompt learning methods in handling these dynamic conditions, the authors propose a benchmark called OpenPL. OpenPL simulates open environments by incorporating dynamic class changes, distribution shifts, and co-evolution of both distribution and classes. Additionally, the work introduces a range of metrics for comprehensive evaluation and provides extensive experiments across various methods, with insights derived from the experimental results.

**Strengths:**

1. The approach to simulating an open-world environment is well-reasoned.

2. A variety of metrics are introduced to evaluate different methods effectively.

3. The experiments are extensive, and insights are offered based on the results.

**Weaknesses:**

More detailed descriptions and analysis of the benchmark construction should be provided.

1. For the Emerging New Classes scenario:

(a) Does the selection of base classes influence the performance of different methods, for example, by selecting either easy or difficult base classes?

(b) Is the number of newly added classes the same at each time step t?

2. The parameter t appears frequently and represents different aspects in each scenario. I suggest using different notation to distinguish these parameters clearly.

3. Since the test process is continuous (with changing classes/distributions), a forgetting score should also be applied to measure performance across methods. For example, performance on the original distribution or class set could be used to assess forgetting [a].

[a] Efficient test-time model adaptation without forgetting. ICML 2022

4. Why is the initial prompt set to “XXXX” instead of the widely used “a photo of a” in prompt learning? Would different prompt initializations affect the performance?

5. Most of the methods tested are few-shot prompt learning methods. Would the proposed benchmark also apply to unsupervised prompt tuning methods [b, c] and test-time prompt tuning methods [d, e, f]? The authors are suggested to provide some discussion about this point.

[b] Unsupervised Prompt Learning for Vision-Language Models

[c] UP-DP: Unsupervised Prompt Learning for Data Pre-Selection with Vision-Language Models, NeurIPS 2023

[d] Test-Time Prompt Tuning for Zero-Shot Generalization in Vision-Language Models, NeurIPS 2022

[e] Diverse Data Augmentation with Diffusions for Effective Test-time Prompt Tuning, CVPR 2023

[f] Historical Test-time Prompt Tuning for Vision Foundation Models, NeurIPS 2024

**Questions:**

Please see Weakness for details.

---

> ### Author Response · Authors · 2024-11-28
>
> Thank you for your suggestion. We appreciate your thoughtful and thorough comments on our paper.
>
> **Weakness 1(a):Does the selection of base classes influence the performance of different methods, for example, by selecting either easy or difficult base classes?**
>
> A1(a): Firstly, we acknowledge that the selection of different base classes can influence the performance of various methods. To address this, we utilized three distinct random seeds and reported the variance across multiple trials to ensure robust results. Furthermore, for the random seeds used in our experiments, we explored related text clustering methods, and our findings confirmed that the selected seeds effectively cover base classes of varying levels of difficulty. This approach ensures a balanced representation and strengthens the generalizability of our results.
>
> **Weakness 1(b):Is the number of newly added classes the same at each time step $t$?**
>
> A1(b):Yes, you are correct. The number of newly added classes remains consistent at each time step $t$.
>
> **Weakness 2: The parameter t appears frequently and represents different aspects in each scenario. I suggest using different notation to distinguish these parameters clearly.**
>
> A2: We thank the reviewer for pointing out this issue. We will make adjustments to the notation in our paper and add subscripts to the parameter t to distinguish it under different paradigms.
>
> **Weakness 3: Since the test process is continuous (with changing classes/distributions), a forgetting score should also be applied to measure performance across methods. For example, performance on the original distribution or class set could be used to assess forgetting [a].**
>
> A3: Thank you for the comment, but we cannot fully agree with it. As stated, our benchmarking focuses on prompt learning in the few-shot scenario. In this scenario, trained models do not involve knowledge forgetting. Instead, it encounters both classes or distributions it is "proficient in" and those it is unfamiliar with.
>
> **Weakness 4: Why is the initial prompt set to “XXXX” instead of the widely used “a photo of a” in prompt learning? Would different prompt initializations affect the performance?**
>
> A4: Thank you for your question. Firstly, we believe that the initialization of the prompt has little impact on the model's performance after multiple iterations. As the number of iterations increases, the influence of initialization diminishes. If it had a significant impact, KgCoOp wouldn't need to specifically set a regularization term to push the prompt towards "a photo of a".
>
> Additionally, we set the initial prompt as 'n' occurrences of "X" to standardize the prompt length across models. Setting the initial prompt to something like "a photo of a" could result in some algorithms requiring the prompt length to exceed a certain value on certain datasets. A longer prompt can harm the model's robustness, making it harder to analyze the performance of prompt learning methods effectively.
>
> **Weakness 5: Most of the methods tested are few-shot prompt learning methods. Would the proposed benchmark also apply to unsupervised prompt tuning methods [b, c] and test-time prompt tuning methods [d, e, f]? The authors are suggested to provide some discussion about this point.**
>
> A5:  Thank you for your insightful comments and suggestions. Our work primarily focuses on few-shot prompt learning methods and aims to provide a robust benchmark specifically tailored to this context. While we appreciate the importance of exploring other types of prompt tuning, such as unsupervised and test-time methods, these are beyond the primary scope of our current work.

---

> ### Author Response · Authors · 2024-12-01
>
> Dear Reviewer owVA,
>
> We have responded to your comments in detail. As the discussion period will end in less than two day, we would like to ask whether there are any additional concerns or questions we could address.
>
> Thanks very much for your effort!
>
> Best regards,
>
> Authors

---

### Official Review · Reviewer_3ZDh · 2024-11-01

**Soundness:** 1
**Presentation:** 3
**Contribution:** 1
**Rating:** 3
**Confidence:** 5

**Summary:**

This paper introduces OpenPL, a benchmark designed to evaluate the robustness of prompt learning methods under open-world conditions, such as the introduction of new classes and distribution shifts. Unlike conventional benchmarks, OpenPL attempts to simulate a more realistic scenario with evolving classes and distribution changes that are common in real-world applications. The paper presents various analyses, comparing several prompt learning methods across different datasets.

**Strengths:**

- **In-depth analysis**: OpenPL challenges current prompt learning generalization benchmarks moving beyond fixed-class setups, aiming to better reflect real-world dynamic conditions.
- **Writing**: The writing is clear and the paper easy to follow.

**Weaknesses:**

- **Unclear motivation**: While the motivation of evaluating robustness of prompt learning method on continuous environment is clear, the paper lack describing what would be the expected desired behavior in each introduced setup. Furthermore, the newly introduce metrics lacks explanation and motivation.

- **Limited novelty**: While the proposed benchmark may be of interest it remains a simple refinement of existing ones and the contribution of the paper may be limited for a conference like ICLR. Furthermore, there is

- **Controversial/vague conclusions**: Several questionable observations are made throughout the paper (see questions).

**Questions:**

- **Regarding the evaluation protocoles**:
    - **Emerging new classes protocole**: The fact that the classification performance declines as new classes are introduced seems obvious to me: as the number of labels increases, the classification problem becomes more difficult. Furthermore, while it is difficult to disagree with the claim that no method "consistently maintain optimal performance", we can see that PromptSRC appears to perform within the top two across most datasets. My point is that the observation made is too vague and it is unclear to me what insight the paper provide here.

    - **Varying ratio of classes**:
    In this scenario the number of classes remains constant and only the proportion of base/new classes varies. Why is the performance of CLIP not stable when varying t  on some datasets (like Eurosat) ? Don't you think that the split of classes may have produce classification problems that are uneven in term of difficulty ? You may consider reporting the relative improvement of each method w.r.t CLIP.

    - **Distribution shifts**: I am not sure why the authors claim that "there is no significant improvement across algorithms when addressing the issue of dynamic data distribution shifts". Don't all prompt learning method improve the performance over CLIP on Imagenet variants for each value of t ? The paper could benefit from a more careful discussion of what constitutes “significant” improvement in this context. If the authors intended to imply that prompt learning should slow the rate of performance decline under distribution shifts, I think that it may be a potentially unrealistic thing to expect.

    - **Co-evolution of distribution and class variation**: What are the domain shifted datasets used in figure 4 ? For instance did you use an Imagenet variant for new-classes and shifted instances and Caltech for base classes and unshifted instances ? On what dataset are the prompt learned in this experiment ? I think we cannot expect prompts learned on a specific and fine-grained dataset like Eurosat to generalize to new classes like the ones of Imagenet, this is why the cross-dataset generalization benchmark is usually performed with prompts learned on Imagenet and evaluated on the fine-grained datasets and not the other way around.

- Given the identified weaknesses, do you have any insights or potential solutions for mitigating these issues in prompt learning methods?

 - Other comments:
    - In related works the description of ProDA is not accurate: "ProDA learns output embeddings of textual prompts rather than input embeddings". ProDA learns an ensemble of prompts in the input embeddings space and use a distributional objective over the output space to learn them instead of using the cross-entropy loss for each one of them independently.
    - You may consider adding the following recent works on prompt learning:
        - Mistretta et al "Improving Zero-shot Generalization of Learned Prompts via Unsupervised Knowledge Distillation" (ECCV24)
        - Lafon et al "GalLoP: Learning Global and Local Prompts for Vision-Language Models" (ECCV24).

---

> ### Author Response · Authors · 2024-11-28
>
> **Thank you for your suggestions. Before addressing your concerns, we would like to take a moment to clarify the motivation and contributions of our work.**
>
> Our primary goal is to systematically and comprehensively evaluate the robustness of prompt learning methods in Vision-Language Models (VLMs) within open-world settings. Previous studies in this area have typically been limited to comparing performance in one or two specific scenarios, which often fail to provide meaningful insights into the true robustness of these methods. In contrast, our work extends the evaluation of prompt learning robustness to a series of more complex and dynamic environments.
>
> Rather than focusing solely on simple performance comparisons, we shift the emphasis towards analyzing robustness curves across a range of challenging, real-world scenarios. We also propose a unified framework for evaluating algorithmic robustness under different metrics, which allows us to derive valuable conclusions about the performance and limitations of prompt learning methods in open-world tasks. Through this comprehensive evaluation, we aim to contribute to a deeper understanding of prompt learning methods' capabilities and their potential for adaptation in dynamic environments.
>
> Moreover, the conclusions are also not "obvious". We provide an analysis to highlight the degree of performance degradation across different methods and to evaluate how well prompt learning methods manage this degradation.  We also provide an analysis of the reasons behind the strong performance of some of the currently top-performing algorithms. We believe these discussions are insightful and could promote more research direction of prompt learning such as combining with TTA methods to address the continue changing problems.
>
> Therefore, the novelty and contribution of our paper lies in the new metrics, new experimental settings, comprehensive evaluations, and insightful discussion. We believe these have enough contribution for the community.

---

> ### Author Response · Authors · 2024-11-28
>
> **Question 1: About Emerging new classes protocole**
>
> A1: We agree that the performance decline as new classes are introduced may appear intuitive due to the increased difficulty with more labels. Our intention was not to suggest that this decline is unexpected, but rather to highlight the degree of performance degradation across different methods, especially under dynamic class shifts, and to evaluate how well prompt learning methods manage this degradation. Indeed, in the final discussion, we also provide an analysis of the reasons behind the strong performance of some of the currently top-performing algorithms.
>
> **Question 2: About Varying ratio of classes**
>
> A2: We have conducted multiple rounds of experiments and reported both the mean values and the variance of the results to ensure robustness. While it is possible to perform additional rounds of experimentation to further mitigate the variability in CLIP's performance across different categories, the final conclusions remain unchanged. Additionally, in our analysis of the robustness of prompt learning methods, we have carefully considered the impact of CLIP’s inherent performance fluctuations. We have reported the relative improvements and degradations in performance compared to CLIP to provide a more comprehensive understanding of the effects.
>
> **Question 3: About Distribution shifts**
>
> A3: We would like to clarify that the performance of prompt learning methods varies across different algorithms in response to dynamic distribution shifts. While most prompt learning methods do improve over CLIP on ImageNet variants at each t-value, the improvement may not always be significant in all cases. In the paper, when we say "no significant improvement," we mean that, despite some improvement, the robustness of the methods does not show substantial progress under distribution shifts. Our aim is not to discredit current algorithms, but to highlight the need for designing more robust methods, possibly through approaches like combining with Test-Time Adaptation (TTA).
>
> **Question 4: Co-evolution of distribution and class variation: What are the domain shifted datasets used in figure 4?**
>
> A4: In our experiments, we trained on ImageNet and used samples from ten smaller datasets as sources for unknown classes and distribution shifts. This approach aligns with your point: we cannot expect prompts learned on a specific and fine-grained dataset like Eurosat to generalize to new classes such as those in ImageNet. This is why cross-dataset generalization benchmarks are typically performed with prompts learned on ImageNet and evaluated on more fine-grained datasets, rather than the other way around.
>
> **Question 5: Given the identified weaknesses, do you have any insights or potential solutions for mitigating these issues in prompt learning methods?**
>
> A5: At this stage of our work, we have primarily focused on evaluating the robustness of prompt learning methods in dynamic environments, and we do not yet have concrete solutions to address the weaknesses identified. While we have pointed out areas where current prompt learning methods struggle—such as performance degradation in more complex and open environments—offering specific solutions requires further exploration. The combination of prompt learning methods and Test-Time-Adaptation may be a promising direction.
>
> **Question 6: About the description of ProDA**
>
> A6: Thank you for pointing out the inaccuracy in our description of ProDA. We apologize for the confusion. As you correctly mentioned, ProDA learns an ensemble of prompts in the input embedding space and optimizes a distributional objective over the output space, rather than independently using cross-entropy loss for each prompt. We will revise the description in the related works section to reflect this more accurately.

---

> ### Author Response · Authors · 2024-11-28
>
> **Question 7: Adding recent works on prompt learning:**
>
> A7: We have reviewed GalLoP ("Learning Global and Local Prompts for Vision-Language Models") and conducted some experiments to assess its performance. Due to computational and space limitations, we are only able to present a subset of the results related to GalLoP in the rebuttal. We report the average performance of the algorithm under dynamic distribution shifts and dynamic co-evolution of distribution and class variation across different t-values. However, we believe these results provide useful insights and demonstrate how this method performs in comparison to other prompt learning approaches.
>
>
> | t/Dataset          | ImageNet-R | ImageNet-A | ImageNet-V2 | ImageNet-Sketch |
> |--------------------|------------|------------|-------------|-----------------|
> | t=0.0              | 85.51      | 85.17      | 85.17       | 85.17           |
> | t=0.2              | 81.81      | 76.67      | 67.10       | 46.00           |
> | t=0.4              | 78.52      | 66.33      | 64.05       | 44.09           |
> | t=0.6              | 76.64      | 57.00      | 67.03       | 50.83           |
> | t=0.8              | 74.73      | 46.67      | 63.78       | 49.79           |
> | t=1.0              | 75.18      | 38.83      | 61.62       | 48.50           |
>
> | t/Dataset          | Caltech101 | EuroSAT | FGVCAircraft | Food101 | OxfordFlowers | OxfordPets | StanfordCars | SUN397 | UCF101 |
> |--------------------|------------|---------|--------------|---------|---------------|------------|--------------|--------|--------|
> | t=0.0              | 90.78      | 99.00   | 90.46        | 92.78   | 91.10         | 95.03      | 86.07        | 79.82  | 90.03  |
> | t=0.2              | 92.56      | 67.12   | 81.88        | 94.58   | 87.78         | 90.84      | 85.28        | 79.08  | 86.83  |
> | t=0.4              | 91.44      | 50.37   | 68.76        | 90.28   | 83.82         | 87.36      | 80.90        | 77.05  | 81.16  |
> | t=0.6              | 90.44      | 30.76   | 54.12        | 84.30   | 77.35         | 87.85      | 75.44        | 72.84  | 74.86  |
> | t=0.8              | 90.11      | 24.02   | 38.42        | 80.19   | 69.66         | 87.79      | 65.31        | 67.21  | 66.21  |
> | t=1.0              | 90.11      | 19.80   | 22.09        | 80.21   | 57.76         | 85.25      | 55.76        | 59.25  | 59.72  |

---

> ### Author Response · Authors · 2024-12-01
>
> Dear Reviewer 3ZDh,
>
> We have responded to your comments in detail. As the discussion period will end in less than two day, we would like to ask whether there are any additional concerns or questions we could address.
>
> Thanks very much for your effort!
>
> Best regards,
>
> Authors

---

> > ### Comment · Reviewer_3ZDh · 2024-12-02
> >
> > Thank you for your detailed response. I fortunately I remain not fully convinced that the observation made in this work are enough for publication at a conference such as ICLR. Therefore I will keep my score.

---

### Official Review · Reviewer_o46u · 2024-11-03

**Soundness:** 3
**Presentation:** 3
**Contribution:** 3
**Rating:** 6
**Confidence:** 5

**Summary:**

The authors propose a new benchmark called OpenPL, where they evaluate the performance of existing prompt learning frameworks in diverse and realistic open world environments. They have an interesting observation that there is no one method that performs well across all scenarios.

They introduce several realistic test scenarios: 1. Dynamic class changes; 2. Dynamic Distribution shifts; 3. Dynamic Co-evolution of distribution and class variation and also performance metrics to evaluate them.

**Strengths:**

1. They establish an impressive benchmark which extensively covers VLMs, prompt learning methods, detailed analysis of these methods in the above defined realistic test scenarios.
2. They introduce several performance metrics carefully designed to analyse the performance of  different algorithms in the open world scenarios. These metrics are very intuitive and aptly designed for this problem.
3. The results are presented in a very clear and concise manner.
4. It is a bold and a much needed evaluation of prompt learning methods. The analysis presented could be very insightly for the research community interested in the adapatation of VLMs.

**Weaknesses:**

**Implementation details:**
1.  The descriptions of charactezing difficulty of a test scenario using $t$ while is fairly well defined, it can be done in a more detailed manner.
2. Please describe how the datasets are set up for this problem in more detail. How the classes are split, sample size etc.
3. This being a benchmark paper, the experimental details are equally important to help reproducibility. The provided details do not suffice I believe.
4. Explaining how the problem is setup in each scenario, taking a dataset as an example can really help the readers.
5. The assumption that new class names are known before testing is unrealistic. As this defines the difficulty of the test scenario, if it is known, one can choose simpler methods that can work more realibly in difficult scenarios?

**Metrics:**
1. All the tables report $Acc(0)$. Based on the definition(line 144, 151, 160), $t=0$ corresponds to evaluation on base task with no new classes or distribution shifts or both. Why is $Acc(0)$ reported in all tables. A suggestion, it is more appropriate to report $Acc(1)$ which corresponds to the most severe case in each scenario which can rather explain your case well.
2. EVM is low for CLIP for most cases. So, can we infer that one is better off using Zero-shot CLIP when one doesn't know the difficulty of test scenario apriori?

**Questions:**

1. In  implementation details, it is mentioned that the maximum number of classes and sample size is fixed for all datasets. But, in Section 3.1, line 137, it is mentioned that half the classes from the dataset serve as base classes and the other half as new classes. How is it actually set?
2. In the case of class changes (3.1 and 3.3), the model is trained on base classes and tested on base and new classes. For eg., in ImageNet, the model is trained for 500 classes with 500 text classifiers. The model is tested on 1k classes, with a 1k length text classifier right? Are the names of new classes assumed to be known before testing? That is not a very realistic assumption to have.
3. $t$ characterizes the difficulty of a test scenario as I understand. A model is trained on base classes and just tested on base and new classes(assuming the new class names are known). The authors mention "As t increases, new classes continually emerge while base classes diminish". This defines different test scenarios but not a changing test scenario as I interpret it. The classes do not continually emerge during test time in the problem right? I request the authors to clarify that this is different from Test Time Adaptation setting. Nothing changes continually during test time. They are all just different test scenarios.
4. Clarify what it means as $t$ increases in line 163.
4. In Figure 2, what is the total number of classes fixed to. Is it fixed for all datasets. If so, what if this is changed? A 50 base + 50 new classes is a very different scenario than 500 base + 500 new classes scenario.

---

> ### Author Response · Authors · 2024-11-28
>
> **Weakness 1:  The descriptions of charactezing difficulty of a test scenario using $t$ while is fairly well defined, it can be done in a more detailed manner.**
>
> A1: We thank the reviewer for pointing out this issue. We will make adjustments to the notation in our paper and add subscripts to the parameter $t$ to distinguish it under different paradigms.
>
> **Weakness 2:  Please describe how the datasets are set up for this problem in more detail.**
>
> A2: For the scenario with dynamic class changes, we randomly select half of the classes as base classes and the other half as new classes. We then perform multiple experiments, averaging the results and reporting the variance.
>
> For the dynamic distribution shift, since the number of categories in ImageNet variants may be fewer than in ImageNet itself, we use all the categories from the variant for testing.
>
> For the coupled distribution and class variation, due to the significant differences in the number of classes and sample sizes between different datasets, we randomly sample from each dataset before merging them. This ensures that, during testing, the number of ImageNet categories is comparable to that of smaller datasets, and the sample sizes for each category are similar to those of the smaller categories.
>
> For example, when combining ImageNet and ImageNet-A, we ensure that the categories come from the 200 classes of ImageNet-A, and the number of samples from either ImageNet or ImageNet-A does not exceed 16 per class in the test scenario. If ImageNet and SUN397 are combined, the number of categories from ImageNet in the test scenario will not exceed 50.
>
> **Weakness 3: All the tables report $Acc(0)$. Based on the definition(line 144, 151, 160), $t=0$ corresponds to evaluation on base task with no new classes or distribution shifts or both. Why is $Acc(0)$ reported in all tables. A suggestion, it is more appropriate to report $Acc(1)$ which corresponds to the most severe case in each scenario which can rather explain your case well.**
>
> A4: We report $Acc(0)$ because this metric reflects the adaptability of algorithms to downstream tasks. Many algorithms, such as MaPLe, demonstrate a relatively high $Acc(0)$; however, as seen with MaPLe under classes dynamic changes, their performance degradation in complex environments is significant. This is precisely the aspect we aim to highlight through our analysis.
>
> Additionally, regarding your suggestion to report $Acc(1)$ as a measure of the worst case performance, we have already introduced $WA$ in Table 1 (line 196) as a metric to capture the worst performance of the algorithms. Thank you for raising this point, and we hope this clarifies our motivation and coverage of performance metrics.
>
> **Weakness 4: EVM is low for CLIP for most cases. So, can we infer that one is better off using Zero-shot CLIP when one doesn't know the difficulty of test scenario apriori?**
>
> A5: Indeed, you make a valid point. In situations where the difficulty of the testing environment is unknown, the performance of Zero-shot CLIP tends to be more stable.
>
> **Question 1: In implementation details, it is mentioned that the maximum number of classes and sample size is fixed for all datasets. But, in Section 3.1, line 137, it is mentioned that half the classes from the dataset serve as base classes and the other half as new classes. How is it actually set?**
>
> A6: Thank you for raising this question. We realize that our explanation may have caused some misunderstanding. The statement in the Implementation Details section was intended to clarify that, under the Dynamic Distribution Shifts and Dynamic Co-evolution of Distribution and Class Variation scenarios, we ensure consistency in the maximum number of classes and sample sizes across datasets. For instance, in the Dynamic Distribution Shifts scenario, the maximum number of classes from ImageNet during testing is not 1000 but is instead aligned with the 200 classes of ImageNet-A. We will revise the wording in the paper to avoid such ambiguities in the future. Thank you for pointing this out.
>
> **Question 2:  Are the names of new classes assumed to be known before testing?**
>
> A7: Yes, you are correct on this point. Before testing, the model is aware of all the categories within the label space, as this is a classification task. Without the classification labels, one might need to rely on clustering methods instead. However, the prompt learning methods reported in our benchmark are not yet capable of achieving this. We are indeed exploring the issues you mentioned as part of our ongoing research.

---

> ### Author Response · Authors · 2024-11-28
>
> **Question 3: I request the authors to clarify that this is different from Test Time Adaptation setting. Nothing changes continually during test time. They are all just different test scenarios.**
>
> A8: Thank you for your observation and for pointing this out. You are correct in your understanding: in our setup, the new classes do not emerge dynamically during the test phase. Instead, $t$ characterizes the difficulty of different test scenarios, where the model is trained on base classes and then tested on a mix of base and new classes (with the assumption that the new class names are known). Our mention of "as $t$ increases, new classes continually emerge while base classes diminish" was intended to describe the design of these distinct test scenarios across different t values, not a dynamic or incremental change during the actual test phase. We agree that this is different from the Test Time Adaptation setting and appreciate your suggestion to clarify this distinction in the paper. We will revise the phrasing to avoid any potential ambiguity.
>
> Thank you again for highlighting this important point.
>
> **Question 4: Clarify what it means as $t$ increases in line 163.**
>
> A9: Thank you for your comment. The increase in $t$ reflects a progression in the difficulty of the test scenarios. Specifically, as $t$ increases, each class in the test scenario includes more samples from ImageNet variants and fewer samples from ImageNet.
>
> **Question 5: In Figure 2, what is the total number of classes fixed to. Is it fixed for all datasets. If so, what if this is changed? A 50 base + 50 new classes is a very different scenario than 500 base + 500 new classes scenario.**
>
> A10: Thank you for your question. To clarify, using ImageNet as an example: in the simplest scenario, when $t=0$, our test scenario consists only of 500 base classes. However, when $t=0.2$, the scenario includes both 500 base classes and 100 new classes simultaneously.

---

> ### Author Response · Authors · 2024-12-01
>
> Dear Reviewer o46u,
>
> We have responded to your comments in detail. As the discussion period will end in less than two day, we would like to ask whether there are any additional concerns or questions we could address.
>
> Thanks very much for your effort!
>
> Best regards,
>
> Authors

---

> > ### Comment · Reviewer_o46u · 2024-12-01
> > **Response to Authors**
> >
> > Dear authors,
> >
> > Thank you for the clarifications, which does answer my questions. I do appreciate the rigorous experiments and analysis presented, as I also mentioned in my initial review. The paper could have really done well with a revision, addressing reviewers' concerns. These clarifications are very much necessary to improve the clarity of the paper. While I very much incline for such a work to be out, I really wished the authors had made use of the opportunity of revising the paper, incorporating the suggestions given through this review process. I choose to retain my retaining, which is positive.

---

> > > ### Author Response · Authors · 2024-12-03
> > >
> > > Dear Reviewer o46u,
> > >
> > > Thank you for your thoughtful feedback and for acknowledging the rigorous experiments and analysis presented in our paper. We truly appreciate your recognition of the contribution our work aims to make. We also understand and respect your feedback regarding the revisions, and we want to assure you that we have carefully considered the suggestions from reviewers and revised the manuscript based on the valuable comments provided during the review process.
> > >
> > > We remain committed to further improving the paper to enhance its clarity and impact. Once again, thank you for your time and positive evaluation of our work. We look forward to your continued support and hope to finalize the manuscript to your satisfaction.
> > >
> > > Best regards,
> > >
> > > Authors

---

### Official Review · Reviewer_JRV8 · 2024-11-04

**Soundness:** 2
**Presentation:** 3
**Contribution:** 2
**Rating:** 5
**Confidence:** 4

**Summary:**

This paper proposes OpenPL, a new benchmark for evaluating the robustness of prompt learning methods for Vision-Language Models (VLMs) in evolving environments.
Key contributions of the paper are as follows:

- **Introducing new evaluation paradigms:**
    - Dynamic class changes scenarios (both emerging new classes and varying ratios of new/base classes)
    - Dynamic distribution shifts scenario using ImageNet variants
    - Dynamic co-evolution scenario where both class and distribution changes occur simultaneously

- **Introducing new performance metrics:**
    - Introduces the Dynamic Robustness Curve (DRC) and several metrics and two robustness definitions based on it
    - Metrics include Area Under Curve (AUC), Worst-case Accuracy (WA), Expected Variation Magnitude (EVM), Variation Stability (VS), Positive Area (PA), and Negative Area (NA)

- **Comprehensive Evaluation:**
    - Evaluates 10 prompt learning methods across 11 diverse datasets

Overall, the paper provides a thorough evaluation framework for understanding how prompt learning methods perform in evolving scenarios where both classes and data distributions can change dynamically.

**Strengths:**

The strengths are:

- The evaluation of the prompt learning methods in the paper is comprehensive, covering 10 methods across 11 datasets. This serves as a valuable reference for future work.
- The paper introduces dynamic changes to the evaluation environment of prompt learning methods.
- The evaluation metrics are also comprehensive, aiming to cover different aspects of robustness in prompt learning methods.
- Prior work has been properly referenced.
- Clear diagrams and tables have been provided to give an overall picture of the performance of different methods in an efficient and useful manner.
- The sections follow a smooth, coherent narrative, and the proper ordering builds step by step toward delivering the main goal of the paper.

**Weaknesses:**

- What the name of the paper suggests is not actually what the paper provides. The title says "realistic evaluation"; however, the evaluation seems to be synthetic in the sense that it uses the exact same datasets that are used in prior work such as CoCoOp and only introduces a parameter `t` (which is why I call it synthetic) that determines the portion of new classes/distribution relative to the training class distribution.

- Common terminology in the literature on prompt learning is not respected in this work. For example, many of the main evaluation paradigms introduced in this paper already exist in prior work such as CoCoOp and MaPLe. Base-to-Novel Generalization corresponds to Dynamic Class Change scenarios, Domain Generalization corresponds to Dynamic Distribution Shift scenarios, and Cross-Dataset Evaluation corresponds to Dynamic Co-evolution of Distribution and Class Variation scenarios. The only difference in this paper is the addition of the parameter `t` that determines the degree or proportion of new class/distribution/dataset samples to base class/distribution/dataset samples. Introducing this new terminology—especially in the case of Dynamic Co-evolution of Distribution and Class Variation—rather than using simpler and more familiar phrases like Cross-Dataset Generalization may be confusing.

- One of the main concerns about the paper is its novelty due to the following reasons:
  - As mentioned, the exact same evaluation scenarios exist in prior work except for the absence of parameter `t`.
  - Moreover, the same 11 datasets are used for the evaluation scenarios as in prior work, which is acceptable; however, when the title suggests "realistic evaluation," the authors should provide material to live up to this promise or change the title.
  - The evaluation of prompt learning methods, while a good reference for any future comparisons and work, does not contain any new discoveries about their performance. In other words, most of the observations reported are already known.

- The authors make some strong negative claims about prompt learning methods while providing no strong evidence, and in some cases, the experiments in the paper itself are not consistent with the claims. For example, there is a claim in the abstract stating that "no current prompt learning method is robust to open environments and no meaningful performance improvement is achieved compared to zero-shot performance." However, by looking at Figure 1, we can see that most prompt learning methods show gains compared to the zero-shot case in the emerging new classes paradigm.

- The robustness definitions and some evaluation metrics have not been carefully crafted and contain logical or notation errors. For example, Performance-Gain Robustness is defined as having AUC - AUC_zs ≥ δ_AUC for all `t`. However, AUC is the area under the curve, making it no longer a function of `t`. The same issue applies to the definition of Decay-Gain-Ratio Robustness.

- There is inaccurate information in the Introduction and Related Work sections about prior research. For example, this part of the paper states, "MaPLe (Khattak et al. (2023a)) proposes benchmarks for Cross-Dataset Evaluation and Domain Generalization by training on ImageNet and altering the test data distribution," suggesting that MaPLe introduces the benchmark. However, the benchmark already exists in CoCoOp, which is an earlier work.

- There are numerous English grammar and writing errors, even in the title. This is especially evident in the Introduction and Related Works sections; for example, "at the earliest time, CoOp (Zhou et al. (2022b)) explored ..."

**Questions:**

- Can you please explain and provide clear evidence for why you think the introduced paradigms aid in the realistic evaluation of the methods? To me, it seems your benchmark is a synthetic benchmark due to parameter `t`. Perhaps you could also consider revising the title.

- As you mention in the paper, the main purpose of prompt learning is to adapt a Vision-Language Model (VLM) to downstream tasks; in other words, it is a form of fine-tuning on specific datasets. Therefore, it is not expected to show the same level of performance on unseen datasets/classes as it does on the source (training) datasets/classes. In fact, a prompt learning method is considered effective if it improves performance on the source dataset while maintaining, or even slightly improving, the original generalization capacity of the VLM on unseen data. However, it seems that most of the strong negative statements made in the paper are based on the assumption that the primary goal of the prompt learning method is to improve the generalization trend of a VLM as parameter `t` increases. If it fails to do so, it is deemed poor. In contrast, where zero-shot performance drops, I expect the same for the prompt learning method, and I do not expect it to remain the same or, even more strangely, to increase. Is this the case that you have the mentioned assumption? If yes, then I believe this is an improper assumption based on prior work. If no, then I suggest revising the strong negative claims, including those in the abstract, Observations 2 and 3 at the end of the introduction, and elsewhere in the paper.

- Observation 4 at the end of the introduction could be considered a new finding and a contribution; however, there is no strong evidence to support this claim in the paper, aside from a brief mention in section 5.5. Please provide evidence for this and include more detail on why you believe so. More generally, if you can present additional observations similar to Observation 4 that go beyond simply analyzing the performance of different methods and instead identify key features in various methods that enable them to outperform others in a scenario, this would significantly enhance the value of your contributions—provided there is strong evidence and experimentation to back it up.

- Please clarify in the paper what you mean by the Dynamic Co-evolution of Distribution and Class Variation scenario e.g. by mentioning that it measures cross-dataset generalization. This is unclear in the paper. The Dynamic Distribution Shift also needs clarification; for example, what happens when other variants of ImageNet are introduced, and why does this change the distribution?

- Please clarify what `x` refers to in lines 203 and 204 inside the table.

- Please clarify lines 213 and 214 under the rank definition; `m` cannot be both 6 and 6xn at the same time.

- Please revise both robustness definitions. AUC, PA, and NA do not depend on `t`.

- It is unclear what kind of robustness the Decay-Gain-Ratio Robustness metric is supposed to measure; please explain in detail what it actually means.

- It might be beneficial to add grids to the diagrams.

- Please explain why the delta values for both robustness metrics are almost the same numbers in Tables 6 to 9.

- There are a considerable number of vague sentences throughout the paper; please clarify them. I understand this may seem too general, but the number of cases is larger than I can mention individually.

- Please address the English writing mistakes; this is a serious issue throughout the paper. I know this might seem too general, but the number of cases is larger than I can mention one-by-one.

- Please ensure that the information regarding prior work in the Introduction and Related Work sections is accurate and informative, as well as anywhere else in the paper. This is also a serious issue.

---

> ### Author Response · Authors · 2024-11-28
>
> **Question 1: Title of the Paper.**
>
> A1: Thank you for your feedback. The term "realistic evaluation" in the title refers to our aim of assessing model robustness under more challenging, evolving test scenarios, which are closer to real-world situations than traditional static test settings. Moreover, it is noteworthy that various similar papers adopted the term "realistic," such as [1], and they also conduct experiments on benchmark datasets such as ImageNet and synthetic settings to simulate more difficult tasks. We believe this is a widely adopted description in the evaluation and benchmark paper.
>
> [1] Realistic Evaluation of Deep Semi-Supervised Learning Algorithms. NeurIPS 2018.
>
> **Question 2: Common terminology in the literature on prompt learning is not respected in this work.**
>
> A2: Thank you for your valuable feedback. We understand your concern regarding the terminology used in the paper. Our intention in introducing new terms was to provide a more structured framework to evaluate the robustness of prompt learning methods across different types of shifts and challenges.
>
> **Question 3: About novelty.**
>
> A3: Thank you for your detailed feedback. We appreciate the opportunity to address the concerns raised.
>
> - While it is true that our evaluation scenarios share similarities with prior works, we believe the novelty of our paper lies in the way we structure and analyze the robustness of prompt learning methods. The introduction of parameter t allows us to systematically control and vary the difficulty of the test scenarios across multiple metrics, which provides a more unified and comprehensive framework for evaluating robustness compared to existing works.
>
> - While the decline in performance of prompt learning methods in complex environments is quite evident, we believe that prior experimental conclusions do not adequately explain whether prompt learning methods can mitigate the extent of performance degradation. As Reviewer o46u mentioned: "It is a bold and much needed evaluation of prompt learning methods. The analysis presented could be very insightful for the research community interested in the adaptation of VLMs." We consider the observations we present to be important.
>
> - Furthermore, previous experiments have primarily relied on direct performance comparisons. In contrast, by introducing a simple modification, we are able to unify the robustness analysis of prompt learning across different scenarios using a series of evaluation metrics. We believe this approach is meaningful and provides more comprehensive insights into the performance of prompt learning methods under various conditions.
>
> We hope this clarifies our approach and the contributions to our work. If you have further questions or concerns, we would be happy to discuss them.
>
> **Question 4: The authors make some strong negative claims about prompt learning methods while providing no strong evidence, and in some cases, the experiments in the paper itself are not consistent with the claims.**
>
> A4: Thank you for your thoughtful comment.
>
> - First, as you noted, our benchmark extends and complements the experimental setups from prior work. The reason earlier algorithms were evaluated under these open-environment scenarios was to demonstrate that prompt learning can be applied to more open settings and exhibit generalization to unseen classes and distributions. Our work aims to provide a more comprehensive analysis of how well prompt learning methods actually perform under such conditions and whether they are robust. Through this effort, we hope to contribute meaningful insights to the development of the field.
>
> - Second, we are not dismissing the contributions of prompt learning methods in improving performance on source (training) datasets or classes, as these improvements are evident. However, this does not necessarily imply that their performance on unseen data (unseen classes or distributions) meets expectations. Our work focuses on systematically evaluating the robustness of these methods through a series of well-defined metrics and highlighting the challenges they still face in unseen scenarios.
>
> - Finally, our goal is not to expect prompt learning methods to maintain unchanged (or even improved) performance in generalized scenarios. Instead, we aim to see a slower performance degradation and more stable behavior in complex scenarios. This aligns with the potential direction for future work, where the focus could be on reducing performance volatility under challenging conditions.

---

> ### Author Response · Authors · 2024-11-28
>
> **Question 5: About Observation 4**
>
> A5: Thank you for your valuable feedback. We acknowledge that Observation 4, while offering insights into prompt learning methods, lacks the level of evidence and experimentation that would strengthen its contribution. We do not have additional experimental results at this time to provide further backing for this observation; this observation is based on initial trends observed in the current experiments, rather than a definitive conclusion. We view this as an area for further exploration in future work.
>
> **Question 6: About Dynamic Co-evolution of Distribution and Class Variation**
>
> A6: Thank you for your comment. In the Dynamic Distribution Shift paradigm, for example, when mixed with ImageNet-A, the test environment consists of only the 200 classes shared with ImageNet-A. However, within these 200 classes, there are samples from both ImageNet and ImageNet-A distributions. This distribution is unknown and is unified into a series of metrics using the parameter $t$.
>
> In contrast, in the Dynamic Co-evolution of Distribution and Class Variation paradigm, both the distribution and the class set may be unknown. We will clarify these points in the paper for better understanding.
>
> **Question 7: Please clarify what refers to in lines 203 and 204 inside the table.**
>
> A7: First, we sincerely apologize for the incorrect representation of the formula in line 204. We will revise it as follows:
>
> $\int_{t\in D} Acc_{zs}(t)-Acc(t)dt, D=\lbrace x|Acc(t)< Acc_{zs}(t) \rbrace$
>
> In this context, the parameter $t$ represents different levels of difficulty in the test scenarios. These two metrics are designed to capture the total area where the performance of the prompt learning algorithm exceeds or falls below the zero-shot CLIP performance.
>
> **Question 8:  Please revise both robustness definitions. AUC, PA, and NA do not depend on $t$.**
>
> A8: Thank you for your question. While it is true that $t$ does not explicitly appear in the definitions, this does not mean that $t$ plays no role. In fact, the entire robustness analysis curve (DRC) is constructed based on $t$. The key purpose of the analysis is to evaluate performance changes across different test scenarios as $t$ varies. Although metrics like AUC, PA, and NA are not directly dependent on $t$ in their definitions, they are derived through a global analysis of the performance over varying $t$. This approach provides a unified and systematic way to evaluate robustness under diverse test conditions.
>
> **Question 9: It is unclear what kind of robustness the Decay-Gain-Ratio Robustness metric is supposed to measure; please explain in detail what it actually means.**
>
> A9: Thank you for pointing this out. The Decay-Gain-Ratio Robustness metric is designed to measure the actual improvement of prompt learning methods compared to zero-shot performance. Specifically, it provides a way to balance and quantify the trade-off between the areas where the performance of prompt learning methods surpasses that of zero-shot CLIP and the areas where it falls behind. By focusing on these regions of the performance curve, the metric aims to evaluate the net benefit of using prompt learning methods in varying test scenarios.
>
> **Question 10: It might be beneficial to add grids to the diagrams.**
>
> A10: Thank you for the suggestion. Adding grids to the diagrams could indeed make them more readable and allow readers to better interpret the data points and trends. We will incorporate grids into the figures in the revised version of the appendix to enhance clarity and visualization. We appreciate your input.
>
> **Question 11: About the writing mistakes and errors in the Introduction and Related Work sections.**
>
> A11: Thank you for bringing this to our attention. We appreciate your feedback regarding the clarity of our writing and recognize that vague sentences can hinder comprehension.
>
> To address this:
>
> - We will carefully review the entire manuscript to identify and rewrite vague or ambiguous sentences. Our goal is to ensure that all descriptions are precise and easily understandable.
>
> - Particular attention will be paid to the Introduction and Related Work sections.
>
> Thank you for your patience and valuable input.

---

> ### Author Response · Authors · 2024-12-01
>
> Dear Reviewer JRV8,
>
> We have responded to your comments in detail. As the discussion period will end in less than two day, we would like to ask whether there are any additional concerns or questions we could address.
>
> Thanks very much for your effort!
>
> Best regards,
>
> Authors

---

> > ### Comment · Reviewer_JRV8 · 2024-12-02
> > **Reply to Authors**
> >
> > Dear Authors,
> >
> > Thank you very much for your responses.
> >
> > It seems that you have not provided a revised manuscript. As such, I am afraid I cannot assess your work based solely on the replies, as some of my main concerns were related to the text in the manuscript.
> >
> > For example, strong negative claims such as "no current prompt learning method is robust to open environments, and no meaningful performance improvement is achieved compared to the zero-shot performance" are made without evidence to support them. On the contrary, all the diagrams in the manuscript indicate that there is indeed a meaningful performance gain over the zero-shot approach.
> >
> > I believe this manuscript requires a thorough revision and carefully crafted statements.
> >
> > Thank you.

---

> > > ### Author Response · Authors · 2024-12-02
> > >
> > > Dear Reviewer JRV8,
> > >
> > > Thank you very much for your comments. **We want to clarify that "robustness" is not the same as "performance".** We want to point out that the statement "no significant improvement" means that the performance trends of prompt learning methods follow the same pattern as that of the zero-shot version, which indicates that the robustness of these methods is heavily reliant on the robustness of the zero-shot model. In other words, despite some performance improvements, these methods do not demonstrate significant improvement in robustness over the zero-shot baseline.
> > >
> > > It is important to note that we are not dismissing the value of prompt learning, but rather highlighting an important research direction for future work. An ideal robustness improvement would be that, as the performance of the zero-shot model degrades in more challenging environments, the performance of prompt learning methods should degrade less severely. In other words, prompt learning methods should mitigate the performance drop observed in the zero-shot model, ideally improving the model’s robustness in open environments. Therefore, we clarify that the robust prompt learning methods should not only achieve performance improvement over the zero-shot version but also achieve robustness improvement.
> > >
> > > If you feel that our conclusions come across as overly negative, we propose revising the observations as follows to better articulate the above points.
> > >
> > > 1. For dynamic class changes, it is challenging for any prompt learning method to consistently achieve optimal performance across different datasets.
> > >
> > > 2. Though achieving performance improvement, prompt learning methods do not demonstrate a noticeable enhancement in robustness over zero-shot performance under dynamic data distribution shifts.
> > >
> > > 3. In scenarios with dynamic distribution and class changes, the robustness of prompt learning methods relies heavily on the robustness of the zero-shot prediction of CLIP, showing no substantial gain in robustness.
> > >
> > > **Due to current time constraints, we are unable to make further changes to the manuscript at the moment, but we will make sure to address other wording concerns and upload the updated manuscript shortly.**
> > >
> > > Thanks very much for your effort!
> > >
> > > Best regards,
> > >
> > > Authors

---

> > > > ### Comment · Reviewer_JRV8 · 2024-12-02
> > > > **Reply to Authors**
> > > >
> > > > Thank your for you response.
> > > > I totally understand the message that you intended to convey to the reader, i.e. "no robustness improvement over zer-shot". However, the current manuscript conveys the message : "no performance improvement over zer-shot".
> > > > Thus:
> > > > If the message of the paper was "no performance improvement over zer-shot" and the evidence really showed this, then I agree that it was an important message for future development of more capable prompt learning methods and I would raise my score. Nonetheless, this is not the case.

---

> > > > > ### Author Response · Authors · 2024-12-03
> > > > >
> > > > > Thank you for your response. We appreciate your understanding of the message we intended to convey, which focuses on the robustness of prompt learning methods. It seems that some of the previous wording may have led to a misunderstanding regarding the contribution of our paper. We will revise the manuscript to clarify that the conclusion is "no robustness improvement over zero-shot." We believe this observation on the robustness of prompt learning is valuable, as designing robust algorithms is crucial in many practical applications, such as drone reconnaissance or medical diagnostics.

---

### Meta-Review · Area_Chair_kvSa · 2024-12-11

**Metareview:**

The paper proposes OpenPL, which comprehensively benchmarks the robustness of existing prompt learning methods for VLMs. Several new metrics are carefully designed including Dynamic Robustness Curve (DRC), AUC, Expected Variation Magnitude (EVM), and Variation Stability (VS) to assess prompt learning methods comprehensively. The benchmark can facilitate future research in this fields by outlining a critical gap in the existing prompting. However, reviewers noted similarities to prior benchmarks like CoCoOp and MaPLe, with the primary addition being a tunable parameter t to control scenario difficulty. Strong negative conclusions about the robustness of prompt learning methods, such as “no meaningful performance improvement over zero-shot,” are contradicted by the experimental results, which often show gains over zero-shot methods. Despite providing detailed responses during the rebuttal period, the authors did not revise the manuscript to reflect these clarifications. As a result, critical issues raised by reviewers remain unaddressed in the submission.

Ultimately, while the work is a step forward in evaluating prompt learning robustness, its incremental nature and unresolved presentation issues suggest that it requires further refinement before being suitable for publication.

**Additional Comments On Reviewer Discussion:**

- Reviewers (JRV8, 3ZDh) questioned the novelty of OpenPL, pointing out its close resemblance to prior works like CoCoOp and MaPLe. The authors highlighted the introduction of the parameter t for controlling test scenario difficulty and the unified robustness analysis using new metrics. They argued that OpenPL extends previous benchmarks by focusing on robustness trends rather than static performance. Many reviewers remained unconvinced, considering the contributions incremental.
- Multiple reviewers (o46u, owVA) requested clearer descriptions of how datasets and scenarios were constructed, as well as detailed explanations of metrics. The authors provided clarifications during the discussion but did not update the manuscript to include these details.
- Reviewer owVA suggested expanding the benchmark to evaluate unsupervised and test-time tuning methods. The authors acknowledged the suggestion and indicated this as a direction for future work.

The authors provided detailed responses to reviewer concerns, clarifying methodological ambiguities and addressing some concerns about the benchmark’s design. However, no revised manuscript was submitted. While reviewers appreciated the authors’ efforts, many retained their original scores, citing the need for substantial revisions to improve the clarity and novelty of this work. Overall, the concerns outweighed the paper’s strengths, leading to a mixed reception and an overall decision to reject.

---

### Decision · Program_Chairs · 2025-01-22

Reject